# Inducible chromatin priming is associated with the establishment of immunological memory in T cells

Sarah L Bevington[1], Pierre Cauchy[1], Jason Piper[2], Elisabeth Bertrand[3], Naveen Lalli[2], Rebecca C Jarvis[1], Liam Niall Gilding[1], Sascha Ott[2], Constanze Bonifer[1] & Peter N Cockerill[1,*]

## Abstract

**Immunological memory is a defining feature of vertebrate physiology, allowing rapid responses to repeat infections. However, the molecular mechanisms required for its establishment and maintenance remain poorly understood. Here, we demonstrated that the first steps in the acquisition of T-cell memory occurred during the initial activation phase of naïve T cells by an antigenic stimulus. This event initiated extensive chromatin remodeling that reprogrammed immune response genes toward a stably maintained primed state, prior to terminal differentiation. Activation induced the transcription factors NFAT and AP-1 which created thousands of new DNase I-hypersensitive sites (DHSs), enabling ETS-1 and RUNX1 recruitment to previously inaccessible sites. Significantly, these DHSs remained stable long after activation ceased, were preserved following replication, and were maintained in memory-phenotype cells. We show that primed DHSs maintain regions of active chromatin in the vicinity of inducible genes and enhancers that regulate immune responses. We suggest that this priming mechanism may contribute to immunological memory in T cells by facilitating the induction of nearby inducible regulatory elements in previously activated T cells.**

**Keywords** chromatin; epigenetics; gene regulation; immunity; memory T cell
**Subject Categories** Chromatin, Epigenetics, Genomics & Functional Genomics; Immunology
The EMBO Journal (2016) 35: 515–535

## Introduction

Immunological memory plays a crucial role in maintaining the mammalian immune response, enabling a rapid reaction to foreign pathogens when they are re-encountered. Memory T cells ($T_M$) arise as a consequence of previous immune reactions and are primed to efficiently upregulate key response genes when re-challenged, while antigen-inexperienced naive T cells ($T_N$) take much longer to respond to the same stimuli (Rogers *et al*, 2000; Sprent & Surh, 2002). The initial activation of $T_N$ by antigen-presenting cells (APC) involves an interaction of the T-cell receptor (TCR) with a specific antigen bound to the major histocompatibility complex on the APC. This occurs in parallel with additional interactions with co-stimulatory molecules, such as CD28 (Brownlie & Zamoyska, 2013). These responses trigger protein kinase and calcineurin signaling pathways which activate inducible transcription factors (TFs) such as AP-1, NFAT, and NF-κB. These TFs, together with preexisting developmentally regulated TFs such as RUNX1 and ETS-1, are required for the expression of inducible cytokines and cytolytic molecules which function within a complex network of immune cells to eliminate infections. The work of numerous laboratories has led to a detailed understanding of the $Ca^{2+}$-dependent activation of NFAT and the kinase-dependent activation of AP-1 and NF-κB, and the roles they play in creating inducible DNase I-hypersensitive sites (DHSs) and in the regulation of TCR-inducible genes such as *IL2*, *IL3*, and *CSF2* (Hogan *et al*, 2003; Johnson *et al*, 2004). ETS-1 and RUNX1 also play wider important roles throughout T-cell development and activation (Muthusamy *et al*, 1995; Telfer & Rothenberg, 2001; Egawa *et al*, 2007; Hollenhorst *et al*, 2009).

When stimulated for the first time, $T_N$ undergo a protracted process of internal reorganization taking 1–2 days, during which they transform from a quiescent spore-like state to actively proliferating T blast cells ($T_B$) (Zhao *et al*, 1998). During this period, the nucleus undergoes extensive Brg1-dependent chromatin remodeling and increases 5- to 10-fold in volume. Once formed, $T_B$ are capable of undergoing many rounds of proliferation in the absence of further TCR stimulation, using IL-2 as a growth factor. In the absence of TCR signaling, these recently activated $T_B$ cells no longer express the inducible cytokines and cytolytic molecules normally associated with fully activated T cells. However, upon re-stimulation, $T_B$ respond like $T_M$ and rapidly react to TCR signaling (Mirabella *et al*, 2010). For example, we previously showed that in $T_B$ cells, a NFAT-dependent DHS at the inducible *CSF2* enhancer can be formed within just 20 min of stimulation (Johnson *et al*, 2004). *In vivo*, the blast transformation phase is typically followed by further differentiation to effector cells such as Th1 and Th2 cells, according to the type of infection. Following clonal expansion and pathogen

1 Institute of Biomedical Research, College of Medicine and Dentistry, University of Birmingham, Birmingham, UK
2 Warwick Systems Biology Centre, University of Warwick, Coventry, UK
3 Section of Experimental Haematology, Leeds Institute for Molecular Medicine, University of Leeds, Leeds, UK
*Corresponding author. Tel: +44 121 4146841; E-mail: p.n.cockerill@bham.ac.uk

clearance, the effector T-cell population contracts, leaving a small proportion of surviving cells. These cells return to the quiescent state as $T_M$, while retaining some of the properties of $T_B$.

Several studies have compared steady-state levels of chromatin modifications and gene expression and identified differences between $T_N$ and $T_M$ in the balance between Trithorax and Polycomb-regulated chromatin domains of histone methylation (Yamashita *et al*, 2004; Araki *et al*, 2009; Nakayama & Yamashita, 2009; Russ *et al*, 2014; Seumois *et al*, 2014; Crompton *et al*, 2015), in domains of histone acetylation (Araki *et al*, 2008), and in DNA methylation (Hashimoto *et al*, 2013; Komori *et al*, 2015). However, while these domains can correlate with differences in steady-state mRNA levels in resting $T_N$ and $T_M$, their relevance to inducible gene expression remains largely unexplored. Furthermore, no mechanisms have yet been described to account for how these domains are created and maintained.

The molecular mechanisms responsible for the disparity between $T_N$ and $T_M$ remain unclear, as there are no obvious candidates for $T_M$-specific TFs. There is, however, ample evidence that epigenetic mechanisms play a role in maintaining T-cell differentiation and memory (Rothenberg & Zhang, 2012). It has long been known that terminal T-cell differentiation to Th1 or Th2 T cells is accompanied by the gain of DHSs at specific cytokine gene loci (Agarwal & Rao, 1998), in parallel with the induction of Th1- and Th2-specific factors such as TBX21 and GATA-3. However, many of the DHSs acquired during the process of activation and differentiation are shared across different classes of differentiated T cells, and the significance of many of these sites remains unclear. For example, the Th2-specific *Il4* gene is associated with several DHSs in the *Il4/Rad50* locus that are present in both Th1 and Th2 cells, but absent in naïve T cells (Agarwal & Rao, 1998; Fields *et al*, 2004). The *Il10* gene is similarly regulated by both Th2-specific DHSs, and DHSs which are also present in undifferentiated T blast cells or in Th1 cells (Jones & Flavell, 2005). Regulatory T cells (Treg) represent another class of differentiated T cells which acquire a specific set of DHSs that are absent in $T_N$. Differentiation to Treg is driven by the TF FOXP3, but also in this case, many of the Treg-specific DHSs are acquired prior to terminal differentiation to Treg (Samstein *et al*, 2012). Taken together, these studies suggest that T-cell activation may create a permissive state at many loci that renders them receptive to any of the various subsequent differentiation-inducing signals they may encounter which are mediated by lineage-specific factors that include FOXP3, TBX21, and GATA-3.

In our previous studies of the human *IL3/CSF2* locus, we investigated the properties of regulatory elements that control the activation of these two highly inducible cytokine genes in T cells. Similar to the above studies, we identified two distinct classes of DHS that were acquired at different stages of T-cell differentiation and activation (Mirabella *et al*, 2010; Baxter *et al*, 2012). One class existed as stably maintained DHSs that were absent in the thymus and in naïve T cells, and were formed during the process of T blast cell transformation, when T cells become activated via TCR signaling for the first time. Once formed, these primed DHS (pDHSs) were preserved for many cell cycles in the absence of continued TCR signaling. Furthermore, the pDHSs in this locus were detected in circulating human memory T cells, suggesting that they are also temporally stable for what might be months or years (Mirabella *et al*, 2010). However, the function of these pDHSs remained unclear as they had

no detectable enhancer activity when measured by transient transfection assays using reporter genes (Hawwari *et al*, 2002; Baxter *et al*, 2012). In addition to these pDHSs, we also identified several inducible DHSs (iDHSs) that appeared transiently in direct response to activation of TCR signaling (Cockerill, 2004; Baxter *et al*, 2012). Similar to iDHSs detected in other inducible genes in T cells, such as *Il4* (Fields *et al*, 2004) and *Il10* (Jones & Flavell, 2005), the iDHSs in the *IL3/CSF2* locus were associated with strong inducible enhancer function in both transient transfection assays and transgenic mice (Cockerill *et al*, 1999; Baxter *et al*, 2012).

In the current study, our aim was to globally identify and characterize the regulatory elements and factors that maintain alternate chromatin states and underpin the recall response characteristic of both $T_M$ and $T_B$. Our aim was to identify common mechanisms that might be shared across all classes of T cells to establish a permissive state that facilitates secondary responses in previously activated T cells. We performed genomewide sequencing of DHSs (DNase-Seq) and chromatin immunoprecipitation assays (ChIP-Seq) of TFs and histone modifications in $T_N$, $T_M$, and $T_B$ before and after stimulation. We identified over 2,000 pDHSs which were absent in $T_N$ but present in both $T_M$ and $T_B$ and were shared by CD4$^+$ T helper cells and CD8$^+$ cytotoxic T cells. In actively dividing $T_B$ cells, many of these pDHSs were stably bound by ETS-1 and RUNX1, which we propose function to maintain chromatin priming. Upon re-stimulation, they were also bound by AP-1 which we propose is required to both establish priming and promote subsequent inducible responses. Significantly, the pDHSs were embedded within islands of active chromatin marked by H3K4me2 and H3K27ac which often encompassed inducible enhancers that could be activated in $T_B$, but not in $T_N$. In contrast to some genes which are silenced in naïve T cells by the Polycomb-regulated H3K27me3 modification, the pDHSs we identified were not marked by H3K27me3 in naïve T cells.

We therefore propose a model whereby many inducible immune response genes exist within inaccessible chromatin in $T_N$ cells until they become primed toward an active chromatin state that is permissive for rapid enhancer induction by NFAT and AP-1, and accessible to additional differentiation-inducing TFs. Our model provides a simple chromatin-based mechanism for maintaining the priming of $T_M$-specific regulatory elements using preexisting TFs which might allow T cells to retain immunological memory without the need for additional specific factors to silence genes in $T_N$.

## Results

To define the regulatory network establishing immunological memory, we performed an integrated set of genome-wide analyses to identify the DNA elements, TFs, co-activators, and histone modifications that correlate with a stable reprogrammed state in $T_M$ and $T_B$. Because we wanted to investigate global mechanisms of T-cell regulation, we chose to examine naturally arising memory-phenotype cells ($T_M$) instead of TCR-specific T cells that may be restricted to a single T-cell lineage and differentiation state. We first purified different populations of CD4$^+$ T helper cells and CD8$^+$ cytotoxic T cells from mice and then analyzed their responses to stimulation of TCR signaling pathways. For this purpose, we used the C42 transgenic mouse model which contains the intact human *IL3/CSF2* locus encoding IL-3 and GM-CSF (Fig 1A) (Mirabella *et al*, 2010; Baxter *et al*, 2012).

**Figure 1. Previously activated T cells stably maintain an extensive array of DHSs at the *IL3/CSF2* locus.**

A    The 130-kb human *IL3/CSF2* BAC transgene. The insulator (Ins) and enhancer elements (E) are shown as boxes.
B    Steps in the route to T blast cell transformation and re-activation. Purified CD4$^+$ or CD8$^+$ T cells were activated with 2 μg/ml ConA for 40 h and then maintained in IL-2 as $T_B$. Cells were re-stimulated with 20 ng/ml PMA and 2 μM calcium ionophore (PMA/I).
C    Inducible mRNA expression levels in CD4 $T_N$ and $T_B$ stimulated with PMA/I for the indicated times. mRNA levels were expressed relative to the levels of beta-2 microglobulin (*B2m*) with SEM. The number of replicates (*n*) for each is shown underneath.
D    UCSC genome browser shot of the human *IL3/CSF2* locus showing DNase-Seq and ChIP-Seq in CD4 $T_N$ and $T_B$ with (red) and without stimulation (black) with PMA/I for 2 h, plus the ENCODE Jurkat T-cell DNase-Seq data (Thurman *et al*, 2012). Black arrows represent stable DHSs and red arrows are inducible DHSs, with the distances in kilobases of the DHS from either the *IL3* or *CSF2* promoters.
E    Human *CSF2* mRNA expression in CD4 $T_N$ and CD4 $T_M$ stimulated for 2 h with PMA/I, expressed as in (C).
F    Southern blot DNA hybridization analyses of DHSs in human $T_N$ and $T_M$ and C42 $T_B$.
G, H    Luciferase reporter gene assays in stimulated Jurkat T cells transfected with the pXPG plasmid containing the *CSF2* (G) or *IL3* (H) promoter alone or in combination with the indicated pDHS and enhancer DNA regions as defined in (D). The *IL3* −4.1/1.5 construct contains a contiguous region spanning the *IL3* promoter from −4.3 kb to +50 bp. The number of replicates (*n*) is shown below, and the error bars indicate SEM (G) or SD (H).

*IL3* and *CSF2* are prototypical cytokine genes that are efficiently induced in $T_M$ or $T_B$, but not in $T_N$ or thymocytes (Mirabella *et al*, 2010). This locus has been intensively studied as a model for defining inducible mechanisms of enhancer activation in $T_B$ (Johnson *et al*, 2004; Bert *et al*, 2007; Baxter *et al*, 2012). Furthermore, this locus contains DHSs which are already known to exist in $T_B$ and $T_M$, but not in thymocytes or $T_N$ (Mirabella *et al*, 2010; Baxter *et al*, 2012), providing obvious candidates for a study of DNA elements that maintain memory and at the same time allowed us to examine whether the actual mechanisms of memory maintenance were conserved between human and mouse.

We prepared subsets of resting T lymphocytes from C42 mouse spleens and further purified $T_N$ and $T_M$. Actively proliferating $T_B$ were prepared by stimulation of CD4 and CD8 T cells for 2 days with the lectin concanavalin A (ConA) to activate surface receptors, followed by a period of rapid proliferation for 2–3 days in the presence of IL-2 (Fig 1B). We confirmed that the process of ConA stimulation was sufficient to rapidly induce the AP-1 family gene *Fos*. However, the human *CSF2* transgene and the mouse chemokine (C-C motif) ligand1 gene (*Ccl1*) were induced much more slowly over a period of 16 h (Appendix Fig S1A). We also used flow cytometry (FACS) to demonstrate that the bulk of each population of CD4 T cells had the expected properties: (i) $T_N$ were predominantly CD62L$^+$ cells lacking the activation markers CD44 and CD25, (ii) $T_M$ were predominantly memory-phenotype cells expressing CD44 but lacking CD62L and CD25, thereby resembling effector memory T cells, and (iii) $T_B$ predominantly expressed the activation markers CD25 and CD44, consistent with activated effector T cells (Appendix Fig S1B).

Each population of T cells was then stimulated for 2 h with PMA and the calcium ionophore A23187 (PMA/I) to activate TCR signaling pathways that induce AP-1 and NFAT activity. By directly activating pathways downstream of the TCR, we ensured that we investigated mechanisms directly associated with epigenetic and transcriptional responses and not just differences in the expression or function of molecules mediating signaling from the cell surface. Analyses of mRNA expression for human *IL3* and *CSF2*, as well as mouse *Il4* and *Il10*, confirmed that each gene was rapidly and strongly induced by PMA/I in CD4 $T_B$ but not in $T_N$ cells (Fig 1C). These experiments verified that inducible cytokine gene loci in $T_N$ and $T_B$ stimulated by PMA/I provide a meaningful model for studying the memory recall response. Both cell types expressed the mRNA for NFAT and AP-1 family proteins at comparable levels prior to stimulation. Following induction with PMA/I over a 2-h time course, both $T_B$ and $T_N$ induced the mRNA to a similar level

albeit with slightly different kinetics (Fig EV1A). This demonstrated that it was not just a difference in TF mRNA expression that distinguishes the responses of $T_B$ and $T_M$ cells from $T_N$, but a differential usage or processing of such factors.

**The human IL-3/GM-CSF gene loci display specific DHSs in $T_M$ and $T_B$**

To map all potentially active *cis*-regulatory elements in T cells from C42 mice, we performed global DNase-Seq which identifies accessible regions of chromatin bound by transcription factors (Cockerill, 2011). Specific analyses of the region covered by the human *IL3/CSF2* transgene detected all of the DHSs previously defined by conventional assays (Baxter *et al*, 2012), plus a previously unknown cluster of DHSs downstream of *CSF2* (Fig 1D). Many of these DHSs had properties consistent with the class of regulatory element defined above as pDHSs. The *IL3* −1.5-kb, −4.1-kb, −34-kb, and −41-kb pDHSs, and the *CSF2* +30-kb DHS were all present in $T_B$ and not in $T_N$. *CSF2* mRNA was also highly inducible in both $T_B$ and $T_M$ but not in $T_N$ (Fig 1C and E). Furthermore, both the *CSF2* +30-kb and *IL3* −34-kb pDHSs were present in circulating human peripheral blood CD4$^+$ CD45RA$^-$ memory-phenotype T cells, at a level indistinguishable from C42 mouse $T_B$, and were weak or absent in human CD4$^+$ CD45RA$^+$ naive T cells (Fig 1F). These findings suggest that pDHSs are (i) maintained in actively proliferating $T_B$ in the absence of TCR signaling, and (ii) contribute to the long-term maintenance of memory in non-dividing circulating $T_M$. These analyses also confirmed that preparation of $T_B$ by stimulation with ConA gave an identical pattern of DHSs in the human *IL3/CSF2* locus to that seen previously for $T_B$ prepared by specific stimulation of the TCR complex and CD28 (Baxter *et al*, 2012), thereby validating the choice of our *in vitro* model.

To further define the features of the stably maintained pDHSs, we performed ChIP-Seq assays for histone H3K4me2 and H3K27ac in CD4 $T_B$. These active chromatin modifications are routinely used to map active enhancer elements (Rivera & Ren, 2013). Within the transgene, pDHSs resided within broad domains marked by both H3K4me2 and H3K27ac specifically in $T_B$ (Fig 1D). In stimulated CD4 $T_B$ ($T_B$$^+$), we detected additional inducible DHSs (iDHSs) which included the well-defined −37-kb and −4.5-kb *IL3* enhancers and the *IL3* promoter, and novel iDHSs 25 and 28 kb downstream of *CSF2*. Significantly, these iDHSs existed in close proximity to pDHSs and zones of active chromatin modifications. Each of these iDHSs, plus the iDHS at the −3-kb *CSF2* enhancer, was highly inducible in $T_B$ but

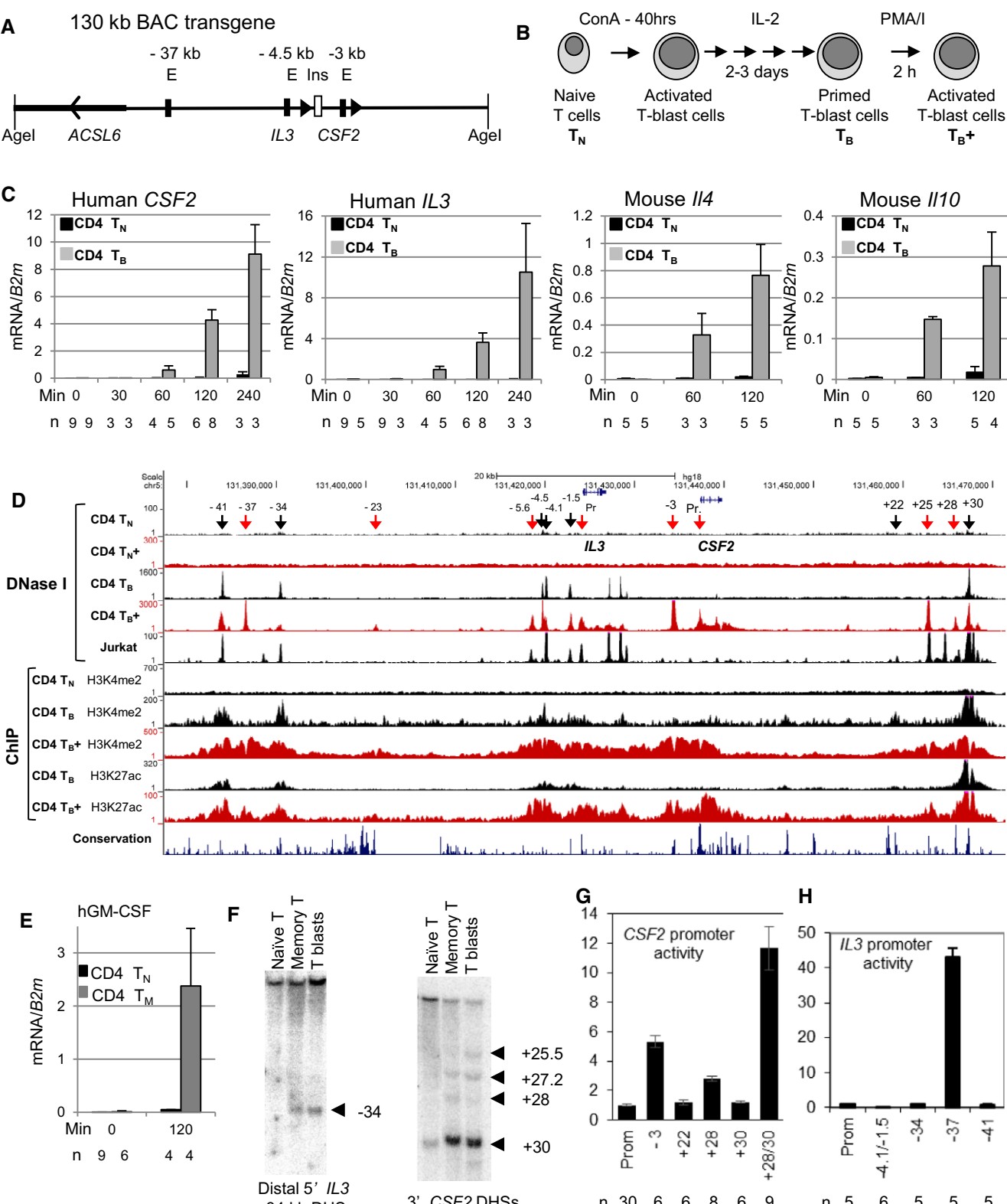

**Figure 1.**

not in $T_N$, consistent with a mechanism whereby the stable pDHSs function to maintain chromatin spanning inducible enhancers in a primed state that allows their rapid response to TCR signaling.

We have previously shown that the −4.1-kb, −34-kb, and −41-kb pDHSs in the *IL3* locus lack enhancer activity, whereas the −37-kb iDHS is a powerful inducible enhancer (Hawwari *et al*, 2002; Baxter

*et al*, 2012). We therefore hypothesized that the human *IL3* −34-kb and −41-kb DHSs were representative of a distinct class of regulatory element that maintain stable zones of active chromatin, but lack classical enhancer activity (Baxter *et al*, 2012). Consistent with this idea, the *CSF2* +22-kb and +30-kb pDHSs also lacked independent enhancer activity when linked to the *CSF2* promoter and tested in reporter gene assays in stimulated Jurkat T cells (Fig 1G), while in the same assay the +28-kb iDHS functioned as a classical inducible enhancer which increased *CSF2* promoter activity threefold. However, although it was inactive on its own, the +30-kb pDHS synergized with the +28-kb enhancer in the activation of the promoter. In addition, a DNA segment that contains both the *IL3* −1.5-kb and −4.1-kb pDHSs similarly lacked enhancer activity in the context of the *IL3* promoter, just like the *IL3* −34-kb and −41-kb pDHSs, and in marked contrast to the −37-kb iDHS (Fig 1H). Taken together, these data suggested that the stable pDHSs are not necessarily classical enhancers, but may aid the induction of inducible enhancers in a natural context when T cells are re-stimulated.

**Activation of the human *IL3* locus is assisted by locus priming**

Human Jurkat T cells display a pattern of DHSs at the *IL3* locus that closely resembles the state seen in $T_B$ (Fig 1D). To investigate the potential function of pDHSs in the absence of classical enhancer activity, we used CRISPR technology to specifically delete the human *IL3* −34-kb pDHS from both alleles in Jurkat cells (Fig 2A). Analyses of average mRNA levels revealed a substantial delay in the activation of *IL3*, but not *JUN*, in 2 clones lacking the −34-kb pDHS when compared to clones with an intact locus (Fig 2B and C). We next employed a PCR-based chromatin accessibility assay to demonstrate that the adjacent −37-kb *IL3* enhancer iDHS had reduced accessibility to DNase I in the clones where the −34-kb pDHS was deleted, whereas no consistent differences were observed in the active TBP promoter or in an inactive control region (Fig 2D). These data are consistent with pDHSs functioning to facilitate inducible enhancer activation by establishing permissive active chromatin domains.

**The human and mouse IL-3/GM-CSF gene loci share conserved pDHSs**

To investigate the formation of pDHSs on a global scale, we used the DNase-Seq data to map DHSs throughout the mouse genome. Figure EV1B shows a comparison of CD4 $T_N$, $T_B$, and $T_M$ and CD8 $T_N$ and $T_B$ across a 900-kb region of the mouse genome which includes the *Il3/Csf2* locus, the *Il4/Il13/Il5* Th2 cytokine gene cluster, plus several other unrelated genes. Shared DHSs present in all samples served as useful internal controls to confirm that appropriate scales were depicted and that equivalent levels of DNase digestion were employed. Most DHSs were conserved between the human and mouse *Il3/Csf2* loci (Figs 1D and EV1B). Most importantly, we detected the same general pattern of pDHSs and iDHSs as seen in the human locus in both CD4 $T_M$, CD4, CD8 $T_B$ and $T_N$, confirming the existence of a distinct class of $T_B/T_M$-specific pDHSs that are stably maintained by memory-phenotype cells. This pattern was highly reproducible because it was seen in 2 independent analyses of CD4 $T_N$ and $T_B$ (Fig EV1B). The 900-kb window, shown in Fig EV1B, plus many other ubiquitously active loci, was used to

set all browser tracks to equivalent levels of detection. The scales used here, and in all the browser tracks depicted below, are shown in Appendix Fig S2. Shown underneath are Venn diagrams depicting overlaps between DHS peak sets detected for the most prominent peaks for the three datasets where biological replicates were available.

Previous studies observed that the promoters of a subset of memory T-cell-specific genes were repressed by the Polycomb-regulated modification H3K27me3 in naïve T cells and thymocytes (Nakayama & Yamashita, 2009). However, this is unlikely to be a universal mechanism, as we showed previously that this was not the case for the promoters of the inducible human *IL3/CSF2* transgene which also cannot be induced in the thymus (Mirabella *et al*, 2010). Consistent with this finding, our analyses of additional published datasets (Wei *et al*, 2009; Russ *et al*, 2014) confirmed that regions spanning the above $T_B$-specific DHSs in the mouse genome had only background levels of H3K27me3 in both CD4 and CD8 $T_N$, where these loci are inactive, and showed the same pattern as differentiated Th2 cells (Fig EV1B).

**Mouse T cells display thousands of memory T-cell-specific pDHSs**

Our next goal was to identify the full complement of pDHSs and iDHSs in mouse T cells. However, this is not a straightforward task because gene regulatory elements are an extremely diverse population that span a broad continuum of DNA elements with varying degrees of similarity to either pDHSs or iDHSs. In order to define the general properties of pDHSs and iDHSs, we first needed to identify representative groups of the most enriched and most specific DHSs within each subset.

We began by evaluating the different properties of DHSs in CD4 and CD8 $T_N$ and $T_B$, and in CD4 $T_M$ to identify the most specific DHSs within each population. To this end, the $T_M$ and $T_B$ DHS datasets were each separately ranked according to fold change of DNase-Seq tag counts first for (i) CD4 $T_M$ relative to CD4 $T_N$ (Fig 3A) and then for (ii) CD4 $T_B$ relative to CD4 $T_N$ (Fig 3B), and data were plotted as density maps spanning a 2-kb window centered on the peak summits. In Fig 3B, the $T_M$ data was also plotted directly alongside the coordinates of the $T_B/T_N$ comparison. Two observations were noteworthy: (i) $T_B$ and $T_M$ shared many specific DHSs that were absent in $T_N$, highlighting the similarity between the recently activated proliferating $T_B$ and the quiescent primary $T_M$; (ii) we reproducibly observed global enrichment of the same population of DHSs in $T_B$ compared to $T_N$ for both CD4 and CD8 T cells when all the data, including independent replicates of the CD4 $T_N$ and $T_B$ samples, were plotted using the same ranking as used for the original CD4 $T_N$ and $T_B$ data shown in Fig 3B (Fig EV2A). Overall, these data suggest that CD4$^+$ T helper cells and CD8$^+$ cytotoxic T cells utilize a common set of DHSs to maintain epigenetic priming in previously activated T cells. It is likely that these sites become established soon after cells first become activated, prior to terminal T-cell differentiation to specific subtypes such as Th1 and Th2. The persistence of pDHSs in CD4 $T_M$ raised the interesting possibility that they may serve the additional purpose of enabling the rapid reactivation of inducible genes in memory T cells.

To obtain mechanistic insights into global mechanisms of pDHS formation, we identified the most prominent pDHSs by using the analysis shown in Fig 3A to define peaks that were at least threefold

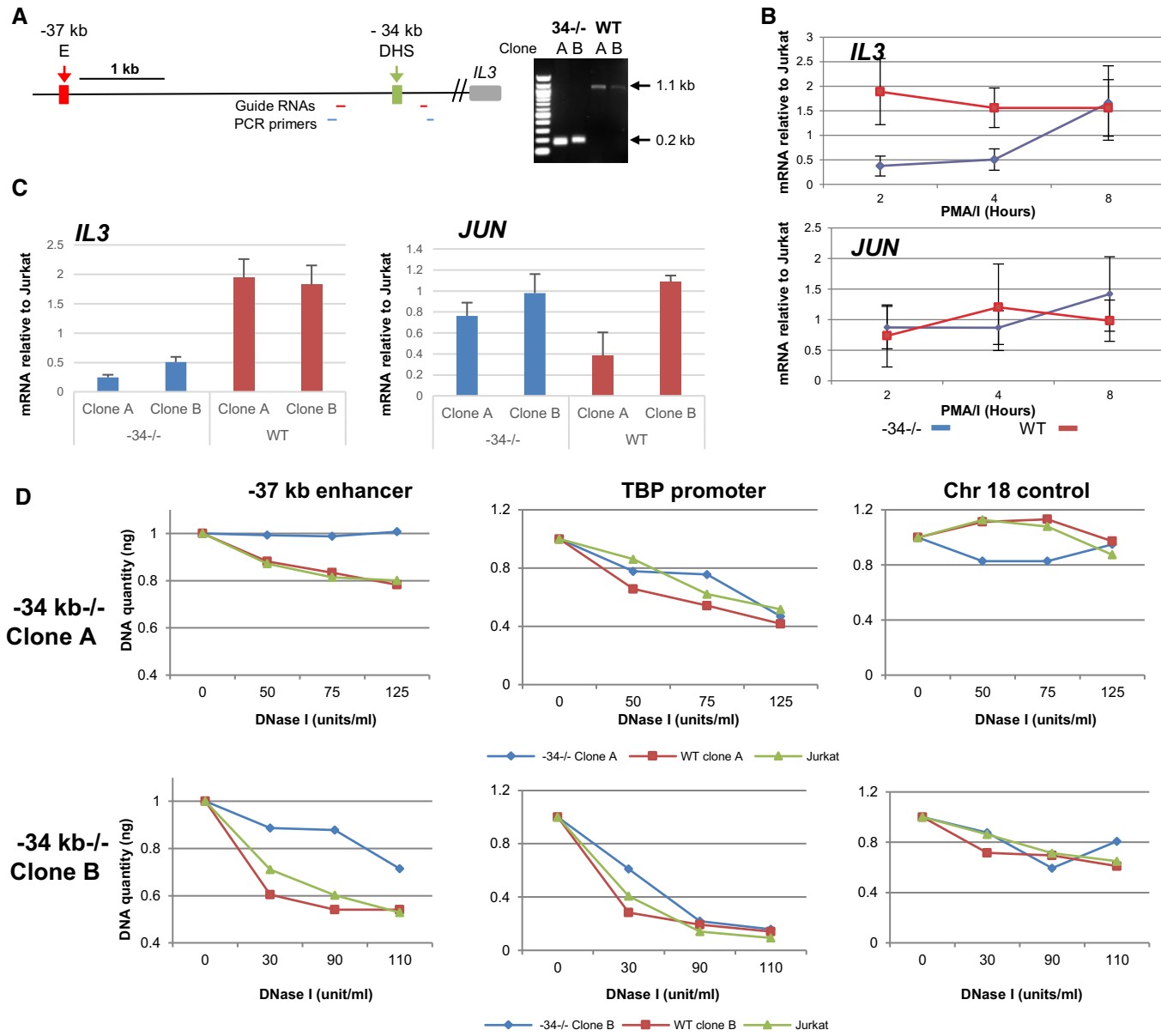

**Figure 2. Impaired induction of *IL3* gene expression and enhancer DHS formation following deletion of a pDHS in human Jurkat T cells.**

A   Map of the region upstream of *IL3* gene spanning the −34-kb pDHS and −37-kb inducible enhancer, together with the locations of the guide RNAs used to delete the −34-kb pDHS and the PCR primers used to detect the deletion. On the right is a PCR analysis confirming deletion of the −34-kb pDHS on both alleles in 2 out of 4 clones selected for the analyses shown below.

B   Average *IL3* (upper) and *JUN* (lower) mRNA expression in the −34-kb$^{−/−}$ clones A and B compared to the WT clones A and B stimulated for 2, 4, and 8 h with PMA/I. mRNA levels were normalized first to *GAPDH* and then to the level of gene expression in untransfected Jurkat T cells. Values represent the average of two −34-kb$^{−/−}$ and two WT clones from two independent experiments (*n* = 4) with SD.

C   *IL3* and *JUN* mRNA expression levels after 2 h of stimulation with PMA/I normalized as in (B). The standard error is shown from five independent experiments.

D   Deletion of the −34-kb pDHS impairs induction of the iDHS at the −37-kb inducible enhancer. The −34-kb$^{−/−}$ clones A and B, the WT clones A and B, and untransfected Jurkat T cells were stimulated with PMA/I for 3 h. A range of DNase I concentrations were used to determine the chromatin accessibility of the −37-kb iDHS in two independent clones, with values expressed relative to normal unstimulated Jurkat cells. Increased accessibility was detected by a reduction in signal detected by qPCR. The active *TBP* promoter and an inactive region on Chr18 are used as controls. Independent experiments for the −34-kb$^{−/−}$ and WT clones A and B compared to the untransfected Jurkat T cells are shown in the upper and lower panels, respectively.

enriched in CD4 T$_M$ relative to CD4 T$_N$. After excluding minor peaks, we selected a reproducible subset of 2,882 of the CD4 T$_M$ pDHSs that were also present in the CD4 T$_B$ DHS dataset (Dataset EV1). The majority of the 2,882 CD4 pDHSs were also present in the CD8

T$_B$ (2,382 = 83%) and the replicate CD4 T$_B$ (85%) datasets shown in Fig EV2A. These 2,882 shared DHSs were then used as a representative, but not necessarily all-encompassing, population of pDHSs for our further analyses. The average DHS profiles for these

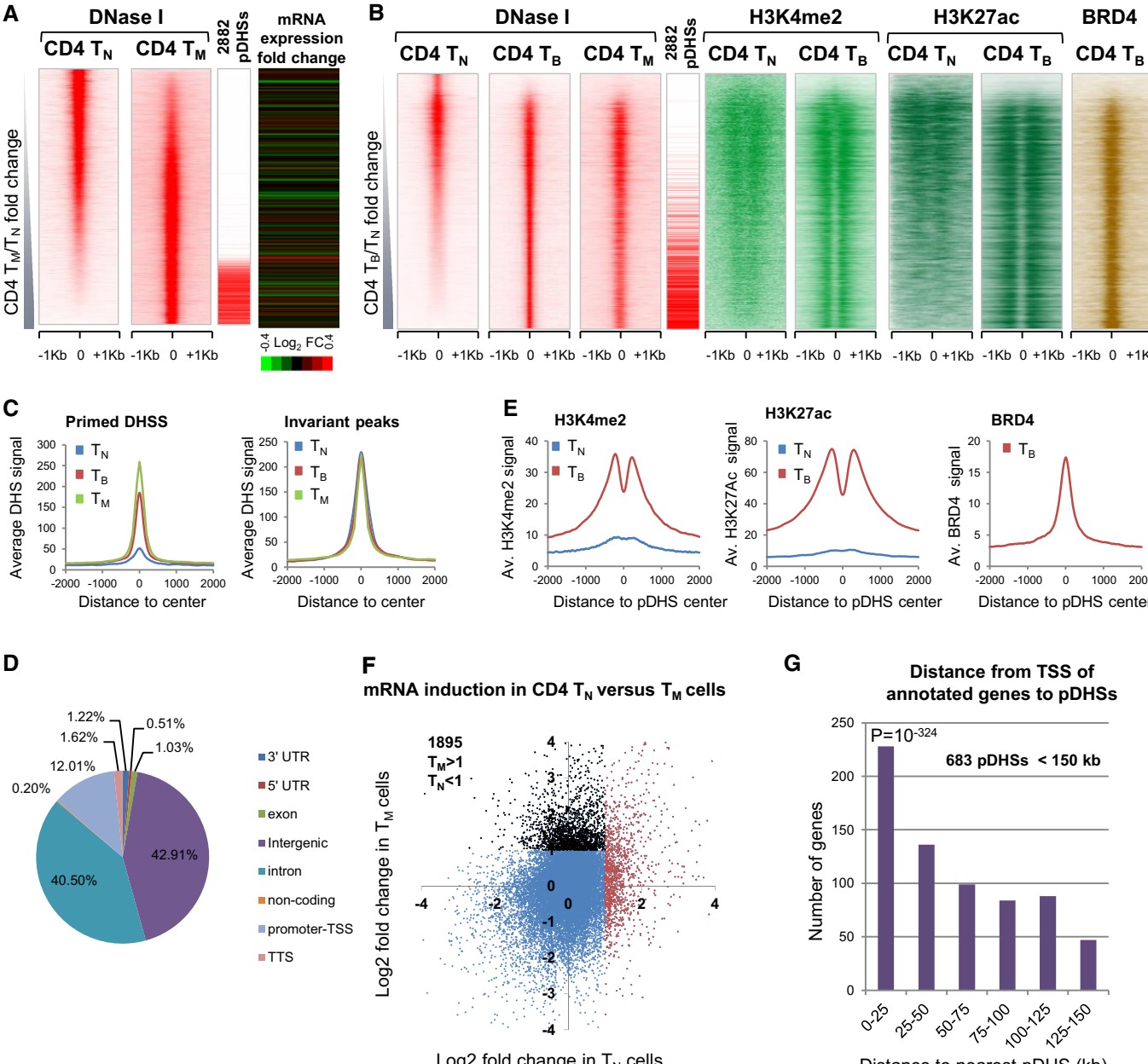

**Figure 3.  Genomewide mapping identifies a class of DHSs restricted to previously activated T cells.**

A   Density maps depicting all DNase-Seq peaks in the order of increasing DNase-Seq tag count signal for CD4 $T_M$ compared to $T_N$. On the right are the locations of the defined subset of 2,882 pDHSs and the log2 $T_M/T_N$ fold change in expression of the closest gene to the corresponding DHS.

B   Density maps for all DNase-Seq and ChIP-Seq peaks shown in order of increasing DNase-Seq tag count signal for CD4 $T_B$ compared to $T_N$. The $T_N$ H3K27ac track is from published data (Lara-Astiaso *et al*, 2014).

C   Average DHS signal at 2,882 pDHSs and 2,882 invariant DHSs in CD4 $T_N$, $T_B$, and $T_M$. The locations of the 2,882 pDHSs are indicated in (A) and (B).

D   Pie chart showing the genomic distribution of pDHSs.

E   Average H3K4me2, H3K27ac, and BRD4 signal at 2,882 pDHSs.

F   Plot of the log2 fold change in gene expression following 2 h PMA/I for CD4 $T_M$ compared to $T_N$. 1,895 genes (black) have a log2 fold change of 1 or above in $T_M$ but not $T_N$.

G   Distance and gene expression analyses. *P*-value represents $\chi^2$ significance against randomly expected number of pDHSs within 25 kb of a TSS (method described in Appendix).

pDHSs in CD4$^+$ cells (Fig 3C) confirmed that they are strong DHSs in both $T_M$ and $T_B$ but weak in $T_N$. This was also true for the average profiles of the 2,882 CD4 pDHSs in CD8 $T_N$ and $T_B$, and in

replicate samples of CD4 $T_N$ and $T_B$ (Fig EV2B). As an internal control for non-specific DHSs, we used a randomly selected subset of 2,882 invariant DHSs which were confirmed as being present in

all three CD4 T-cell types (Fig 3C). The 2,882 pDHSs were mapped back to the genome and consisted mainly of distal sites (83%) in intergenic and intronic regions, with just 12% being within 1 kb of a transcription start site (TSS) (Fig 3D).

We next measured the level of two enhancer-associated histone modifications at all of the DHSs by ChIP-Seq. The enrichment of H3K4me2 and H3K27ac in $T_N$ and $T_B$ was plotted for each of the DHSs as density maps (Fig 3B). As observed for the pDHSs at the human *IL3/CSF2* locus (Fig 1D), the 2,882 pDHSs were marked with H3K4me2 and H3K27ac only in $T_B$ and not $T_N$ (Fig 3B and E). We also performed ChIP-Seq for the transcriptional activator BRD4, which binds to acetylated histones and has been shown to maintain binding during mitosis (Zhao *et al*, 2011), and found a strong enrichment at pDHSs in $T_B$ (Fig 3B and E). In contrast, the average profiles for H3K27me3 taken from published datasets (Wei *et al*, 2009; Russ *et al*, 2014) were essentially baseline across the pDHSs in CD4 and CD8 $T_N$, and in Th2 cells (Fig EV2B), suggesting that these regions are not differentially regulated via loss of a repressive histone modification.

### Activated $T_B$ and $T_M$ express a common set of inducible genes

In parallel with the genomewide DNase-Seq and ChIP-Seq analyses, we measured mRNA expression levels in duplicate in the different CD4 cell types ($T_N$, $T_B$, $T_M$) and in CD8 $T_N$ and $T_B$ by microarray analysis in both untreated cells and cells after 2 h of stimulation with PMA/I. The comparison of the fold induction for each gene in CD4 $T_N$ and $T_B$ showed 1,895 $T_M$-specific genes being at least twofold induced in $T_M$ and less than twofold induced in $T_N$ (black dots, Fig 3F, Dataset EV2). In addition, the average fold change values confirmed that the 1,895 CD4 $T_M$-specific induced genes were upregulated in both CD4 $T_M$ and $T_B$ (Fig EV2C). However, there was little difference in steady-state levels of mRNA among $T_N$, $T_B$, and $T_M$, with values actually being slightly lower in $T_M$ and $T_B$ (Fig EV2D). Gene ontology analysis indicated that these 1,895 upregulated genes were generally associated with immune responses and the activation of signaling pathways (Fig EV2E). In addition to the genes discussed above, this included many immune regulators such as the interleukins 9, 19, 20, 22, 27, and 31, receptors for CSF-1, IL-1, IL-2, IL-12, IL-15, and IL-13, other genes in cytokine or chemokine pathways such as *Cxcl3,* and *Cxcl5, Cxcr3, Cxcr5, Ccl2, Ccl19, Ccl24, Ccr2, Ccr6, Tnfrsf4* and *Tnfrsf9, Tnfsf8,* and genes encoding the transcription factors NFIL3, NFATc1, and NFAT5. Note that although *Nfatc1* was only modestly induced in $T_B$ (Fig EV1A), it was induced more strongly in $T_M$ than in $T_B$ (Fig EV2F).

To search for common patterns of differential mRNA expression among the various T-cell subsets, we performed a clustering analysis of correlation coefficients for the top 1% of genes showing the highest variance between all the subsets (242 genes, Fig EV2G). This revealed a common expression pattern shared by stimulated CD4 $T_M$, CD4 $T_B$, and CD8 $T_B$, whereas the patterns for stimulated $T_N$ cells more closely resembled non-stimulated $T_N$ cells for both CD4$^+$ and CD8$^+$ T cells. Interestingly, the basal pattern for CD4 $T_M$ was more similar to CD4 $T_N$ than it was to the stimulated CD4 $T_M$, likely reflecting the resting proliferation status of these cells. These observations are significant because they suggest that the presence of 2,882 pDHSs has little impact on baseline levels of gene expression, consistent with Figs 3A and EV2D and the notions that (i) the

role of pDHSs is to regulate inducible, not basal gene expression, and (ii) that pDHSs are not classical enhancers that activate genes in the absence of stimulation. A direct comparison of mRNA values for the genes nearest to the 2,882 pDHSs, and for the 1,895 inducible genes identified in Fig 3F, also demonstrated that there was no substantial difference in the expression of these genes in the absence of stimulation in $T_N$ compared to $T_M$ (Appendix Fig S3).

### Primed DHSs are located proximal to inducible genes carrying inducible DHSs

To globally assess the role of pDHSs in regulating inducible gene expression, we mapped the distances from the TSSs of the 1,895 $T_M$-specific inducible genes to the nearest pDHS. We identified 683 genes (Dataset EV3) that had a pDHS located within 150 kb of the TSS of the induced gene, representing 23.7% of all pDHSs (Fig 3G). This figure was higher compared to the number of similarly sized, randomly generated coordinates (214/2,882, 7.4%), randomly selected constitutive DHSs (223/2,882, 7.7%) and randomly selected iDHSs (438/2,882, 15.2%) located within 150 kb of an inducible gene. Remarkably, more than 200 of these 683 pDHSs were located within just 25 kb of the start site ($P \leq 10^{-324}$), indicating a strong correlation between proximity to preexisting pDHSs and inducible gene expression in memory T cells. Of the remaining inducible genes, 91% did nevertheless contain a constitutive DHS within 150 kb.

To identify DNA elements controlling the induction of gene expression, we mapped the inducible DHSs (iDHSs) throughout the genome in CD4 $T_B$ which had been stimulated for 2 h with PMA/I ($T_B^+$). In parallel, we performed ChIP-Seq for H3K4me2, H3K27ac, and BRD4. The two sets of DHS data were plotted as density maps ranked according to the fold change of DNase-Seq tag counts for the combined $T_B^+$ and $T_B$ datasets (Fig 4A). This analysis revealed several thousand iDHSs which were associated with steadily increasing levels of H3K4me2 flanking the iDHSs, and increasing levels of mRNA expression for the adjacent genes, suggesting that many of these elements act as enhancers.

To identify and characterize iDHSs, we initially focused on the top 15% of iDHSs and defined a population of 6,823 iDHSs that were induced by at least 5.5-fold relative to CD4 $T_B$ (Fig 4A). However, since the most highly induced peaks showed the strongest correlation with changes in both H3K4me2 and mRNA expression, we also performed a more stringent selection of the 1,217 strongest iDHSs that were enriched by at least 11-fold (Dataset EV4). The average DHS profiles for these 1,217 highly specific iDHSs confirmed that they were strictly inducible, present only in $T_B^+$ and $T_M^+$ and not in the un-stimulated $T_B$, $T_N$, or $T_M$ (Fig 4B). However, there was also a subset of 1,049 DHSs that were diminished by at least fourfold after stimulation (dDHSs), and the inducible changes in the average H3K27ac profiles of dDHSs suggested both a decrease in the level of H3K27ac and a closing up of the nucleosome-free regions at these sites (Fig 4C). This subset included 249 pDHSs, suggesting that the chromatin encompassing pDHSs and iDHSs is highly dynamic. In contrast, iDHSs became highly enriched for H3K27ac, H3K4me2, and BRD4 specifically after the cells were stimulated (Fig EV3A). Similar to the pDHSs, the 1,217 iDHSs were located mainly in intragenic and intronic regions, with just 1.4% being within 1 kb of a TSS (Fig 4D).

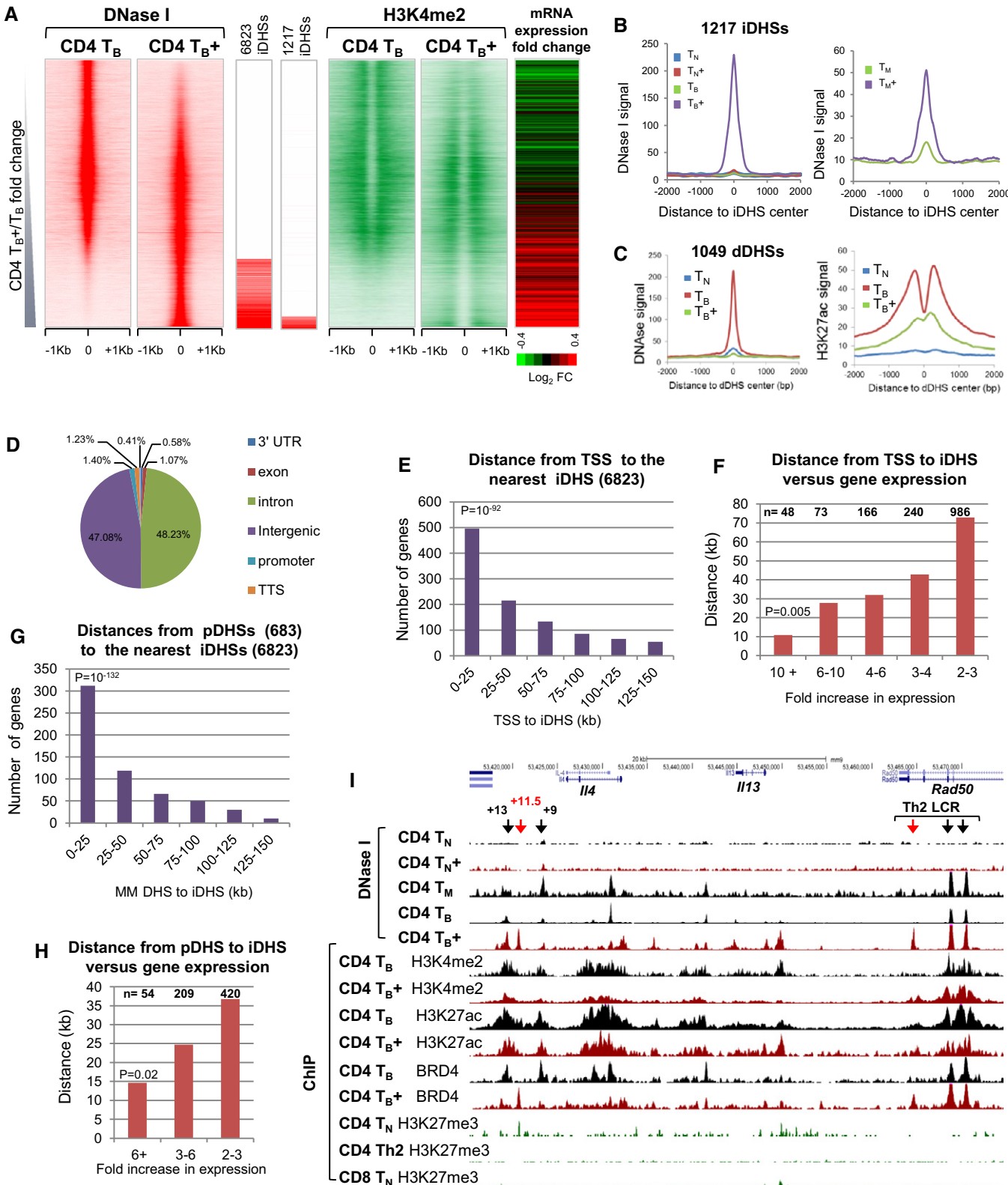

**Figure 4.**

We next investigated the potential significance of the iDHSs with respect to gene expression and measured the distances from the TSSs of the 1,895 $T_M$-specific inducible genes (Fig 3F) to the nearest iDHS (using the more inclusive 6,823 subset). Approximately 1,000 of the inducible genes had an iDHS within 150 kb of the TSS, and nearly 500 of these elements were located within just 25 kb which

◀

**Figure 4.  iDHSs lie close to pDHSs and are associated with inducible genes.**

A   Density maps identifying iDHSs and showing DNase-Seq and H3K4me2 ChIP-Seq peaks in order of increasing DNase-Seq tag count signal for CD4 $T_B^+$ compared to $T_B$ cells. Also depicted are the locations of 6,823 major DHSs that are 5.5-fold induced and a subset of 1,217 of these iDHSs that are 11-fold induced. On the right is the log2 $T_B^+$/$T_B$ fold change in expression of the closest gene to the corresponding DHS.

B   Average DNase I profiles of the 1,217 iDHSs in $T_N$ (+/− PMA/I) and $T_B$ (+/− PMA/I) (left), and in $T_M$ (+/− PMA/I) (right).

C   Average DNase I and H3K27ac profiles of the 1,049 dDHSs in $T_N$ and $T_B$ (+/− PMA/I) which are fourfold diminished after stimulation.

D   The genomic distribution of the 1,217 iDHSs.

E–H   Barplots showing the number of $T_M$-specific genes with an iDHS within 150 kb of the TSS (E); the median distances between the TSSs and the closest iDHS of the $T_M$-specific genes grouped according to the fold induction in $T_M$ after 2 h plus PMA/I compared to $T_M$ for genes which had a TSS < 1 Mb from an iDHS (F); the number of $T_M$-specific genes which have an iDHS within 150 kb of the 683 pDHSs (G); and the median distances from the closest pDHS to the closest iDHS grouped according to the fold induction in $T_M$ after 2 h plus PMA/I compared to $T_M$ (H). *P*-values represent either $\chi^2$ significance against randomly expected number of DHSs within 25 kb (F and H) or *t*-test significance against equally sized random DHSs (E and G). The methods used to calculate *P*-values are described in the Appendix.

I   UCSC genome browser shot of the Th2 *Il4/Il13/Rad50* locus showing DNaseI-Seq and ChIP-Seq. Black and red arrows represent pDHSs and iDHSs, respectively. The values above the arrows indicate the distance in kb from the *Il4* promoter.

was highly significant ($P = 10^{-92}$) (Fig 4E), further signifying a role for the iDHSs as inducible enhancers of these genes. Consistent with this finding, the most highly induced $T_M$-specific genes were associated with the closest iDHSs, whereby the 48 genes that were induced greater than 10-fold were on average within 10 kb of an iDHS (Fig 4F) suggesting a role of these elements as enhancers.

We next addressed the question of the potential relationships between pDHSs, iDHSs, and inducible genes by focusing on the 683 pDHSs which were within 150 kb of a $T_M$-specific inducible gene (Fig 3G) and measuring the distances between these pDHSs and the closest iDHS (Fig 4G). Significantly, 587 (86%) of the 685 pDHSs contained an iDHS within 150 kb, and 312 of these (46%) were within just 25 kb ($P = 10^{-132}$). These data reinforce the view that pDHSs function at close range to support inducible enhancer function. In addition, we found that the most highly induced genes also displayed the shortest distances between the pDHSs and iDHSs (Fig 4H). Hence, the 54 genes that were induced at least sixfold featured a pDHS located within an average of 15 kb of an iDHS. Significantly, 187 of these 312 loci also carried an iDHS that was located within 25 kb of the TSS. Gene ontology analysis of these 187 genes showed a highly significant association with immune responses and the activation of signaling pathways (Fig EV3B) including immune regulators such as *Il3, Il4, Il31, Il15ra, Il1rl1, Il18rap, Cxcr3, Ccr2, Tnfrsf4, Tnfrsf9, Tnfsf8, Tnfaip8, Itga3, Itgav, Alcam, Mpzl2, Lpar3, Notch1, Icos, Map3k5,* and *Fyn,* and key TF genes such as *Rela, Nfil3,* and *Nfatc1*.

To further investigate the potential significance of pDHSs, we plotted their positions relative to iDHSs located within 150 kb of a promoter of an inducible $T_M$-specific gene (Appendix Fig S3C). This confirmed the strong trend for pDHSs to be located close to iDHSs, independently of the distances of these DHSs from the promoter.

To provide additional examples for loci that utilize pDHSs and iDHSs, we examined two other classic models of inducible cytokine loci: the Th2 cytokine gene locus, encompassing *Il4, Il13,* and the Th2 locus control region (LCR) (Lee *et al*, 2005) (Fig 4I), and the *Il10* locus (Jones & Flavell, 2005) (Fig EV3C). Similar to *Il3* and *CSF2, Il4* and *Il10* were specifically induced in $T_B$ and $T_M$ but not $T_N$ (Figs 1C and EV3D) and contain multiple pDHSs and iDHSs which were only detected in previously activated T cells (Figs 4I and EV3C). The Th2 cytokine locus contains several $T_B$-specific iDHSs, including one 11.5 kb downstream of *Il4,* and one within the Th2 LCR, which were each shown previously to function as enhancers (Lee *et al*, 2001). Both of these iDHSs lie close to pDHSs in $T_B$

and $T_M$ (Fig 4I). Like the pDHSs at the *IL3/CSF2* locus (Fig 1D) (Baxter *et al*, 2012), the *Il4* +9-kb pDHS (defined by others as HS4) also lacked independent enhancer function (Lee *et al*, 2001). A similar example is shown for *Il10* (Fig EV3C) where an array of pDHSs and iDHSs extends 30 kb upstream of the promoter specifically in previously activated cells. The *Il4* +11.5-kb and the *Il10* −24-kb iDHSs are both included within primed regions having elevated levels of H3K27ac and H3K4me2 in $T_B$ prior to stimulation. A parallel analysis of published H3K27me3 datasets for *Il4* and *Il10* again showed no prior enrichment for this modification in CD4 or CD8 $T_N$ in regions that subsequently form $T_M$-specific DHS during blast cell transformation (Figs 4I and EV3C). However, there was a $T_N$-specific enrichment for H3K27me3 at the *Il10* promoter, consistent with this being a parallel mechanism of regulating the transition from $T_N$ to $T_B$ and $T_M$, but operating at a different class of regulatory elements.

### Primed DHSs bind constitutively expressed transcription factors

The 2,882 pDHSs created during T-cell blast cell transformation are stably maintained in both dividing and non-dividing cells, without any substantial change to the TF expression program. Therefore, a key question is why the pDHSs remain open in $T_B$ and $T_M$, while the iDHSs are not maintained. To this end, we investigated which transcription factors remained stably bound to pDHSs in the unstimulated state by searching for enriched DNA motifs within the pDHSs using HOMER (Heinz *et al*, 2013). The most enriched motifs corresponded to the known consensus binding sequences for ETS and RUNX family transcription factors, which were present in 44 and 47% of the targets, respectively (Fig 5A). Significantly, the ETS motif identified here corresponds to the composite ETS/RUNX motif to which these factors bind in a cooperative fashion in T cells (Hollenhorst *et al*, 2009). To a lesser extent, pDHSs were also enriched for motifs for inducible factors which respond to TCR and growth factor signaling, including AP-1, IRF, and STAT family proteins. We plotted the coordinates of the motifs back onto the DHSs, ordering them as previously according to the DHS intensity fold change for $T_B$ compared to $T_N$ (Fig 5B). Consistent with their known functions throughout T-cell development, the RUNX and ETS motifs were found both at $T_B$-specific pDHSs and at constitutive DHSs shared by $T_B$ and $T_N$. In contrast, the motifs for AP-1 and STAT were predominantly found within the $T_B$-specific DHSs. We also performed a bootstrapping analysis which determined the statistical significance

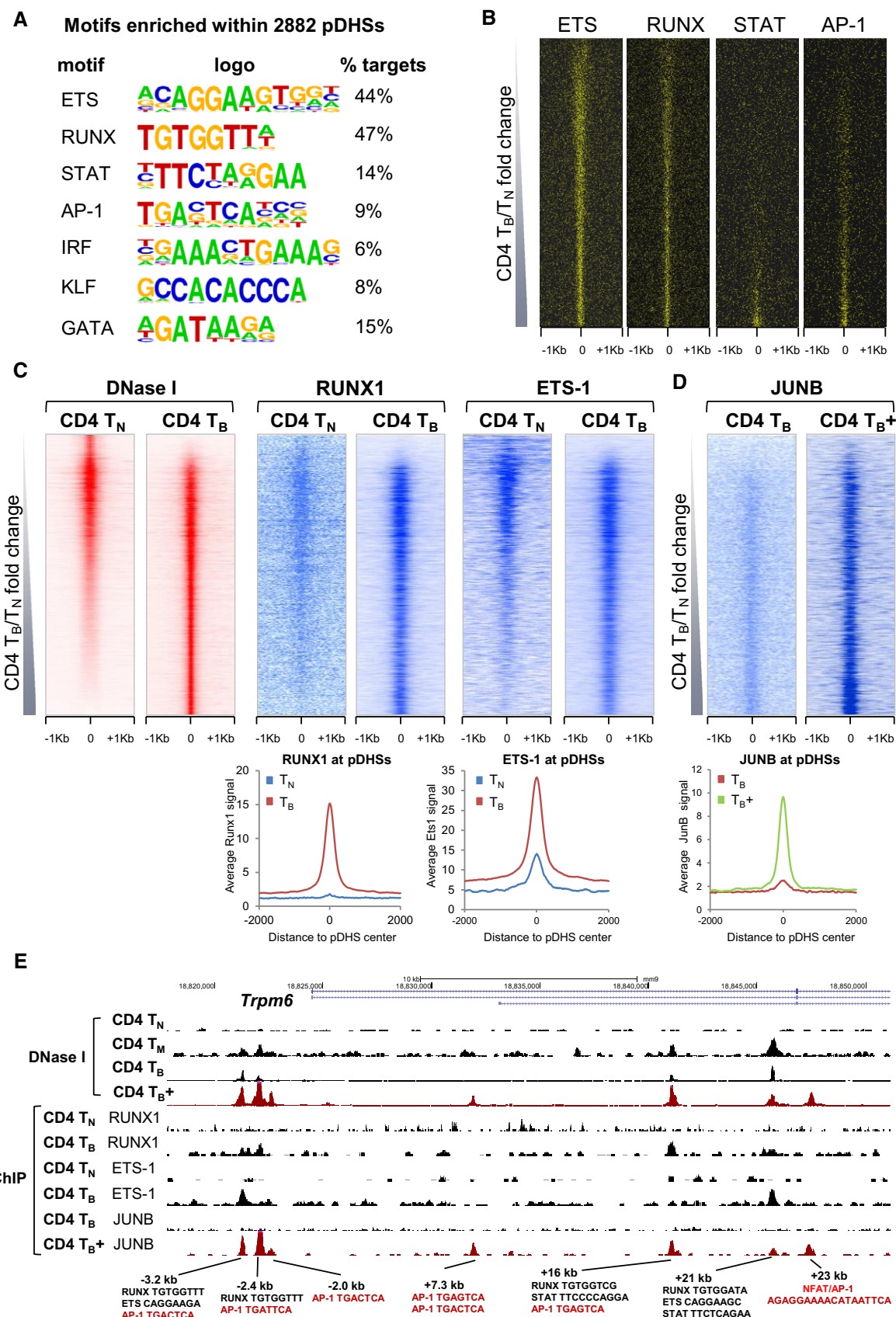

**Figure 5.**

◄

**Figure 5.  pDHSs bind constitutively expressed transcription factors.**

A   *De novo* motifs enriched within 2,882 pDHSs determined using HOMER.

B   Motif distribution in all DHSs ordered by increasing DNase-Seq tag count signal for CD4 T$_B$ relative to T$_N$ as in Fig 3B.

C   DNase-Seq and RUNX1 and ETS-1 ChIP-Seq density maps showing the binding at the DHSs ordered as for (B), with average profiles of RUNX1 and ETS-1 binding to the pDHSs in T$_B$ compared to T$_N$ shown below.

D   JUNB ChIP-Seq in T$_B$ and T$_B^+$ at the DHSs defined in T$_N$ and T$_B$ and ordered as in (B) with average profiles shown below.

E   Patterns of RUNX1, ETS-1, and JUNB binding at RUNX, ETS, and AP-1 motifs at the *Trpm6* locus.

of motif co-occurrence within 50 bp and observed that the ETS, RUNX, STAT, GATA, and AP-1 motifs co-localized within the pDHSs (Fig 6A) and most likely form interacting complexes.

To verify that the predicted TF motifs were occupied *in vivo*, we performed ChIP-Seq to assess the levels of RUNX1 and ETS-1 binding to the DHSs in both T$_N$ and T$_B$ (Fig 5C). RUNX1 and ETS-1 binding was detected in both cell types at the shared DHSs. However, binding of these factors to the pDHSs was restricted to T$_B$ (Fig 5C, Appendix Fig S4). We excluded the idea that this result was caused by increased RUNX1 and ETS-1 expression by examining gene

expression microarray data obtained from independently prepared duplicates for each of the different cell types (Fig 6B) which revealed similar levels of *Runx1* and *Ets1* mRNA in T$_N$, T$_M$, and T$_B$. We therefore hypothesized that additional inducible factors may be required for the original genesis of the pDHSs and investigated the levels of AP-1 binding by performing a JUNB ChIP-Seq assay. Whereas very little basal level AP-1 binding was observed in T$_B$ (Fig 5D), AP-1 binding was greatly increased in T$_B^+$ after stimulation, with high enrichment at the pDHSs, consistent with the elevated AP-1 family mRNA levels (Fig EV1A) and the observed

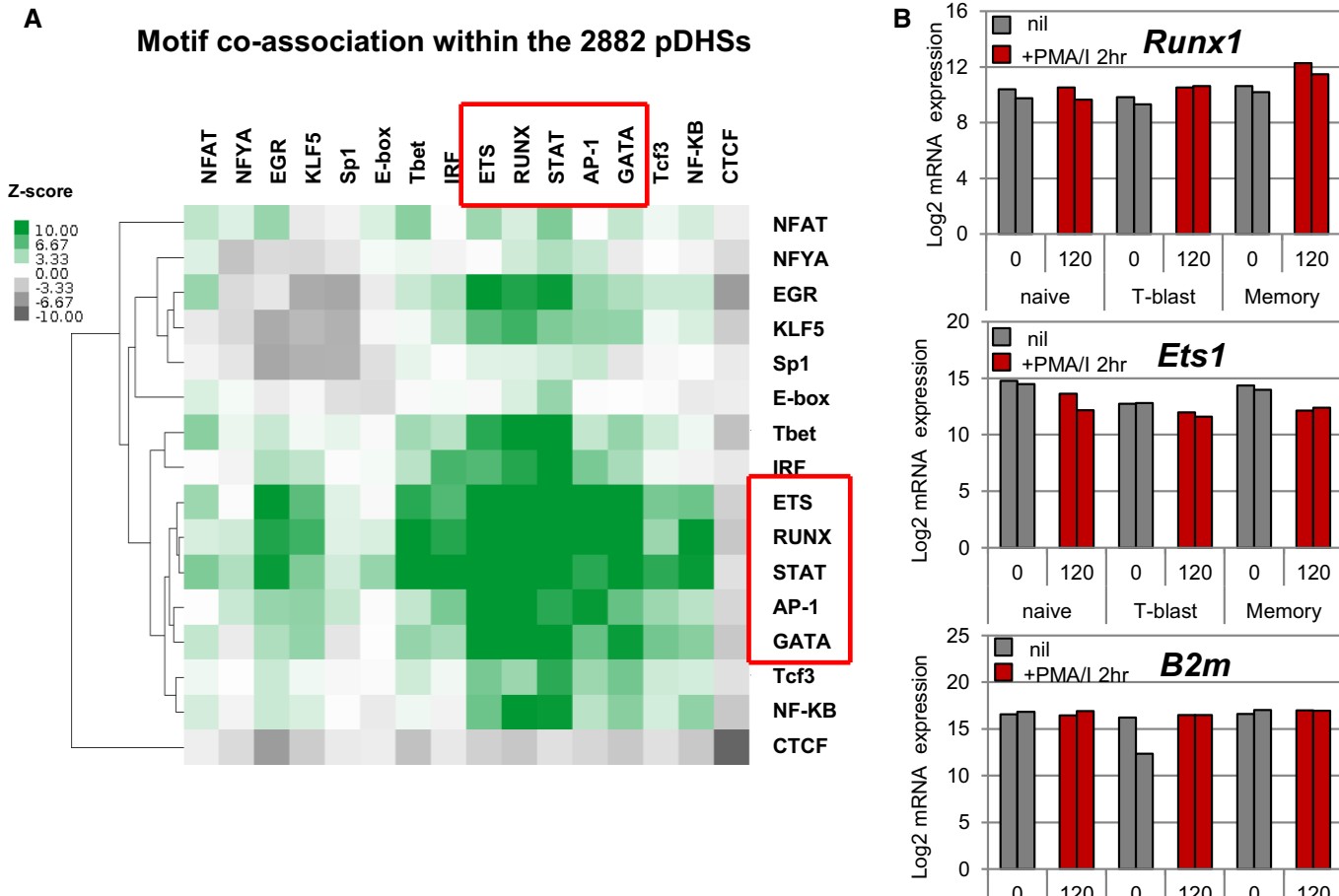

**Figure 6.   Co-association of ETS-1 and RUNX1 in pDHSs, and the role of RUNX1 in establishing priming.**

A   Hierarchical clustering of motif co-association enrichments in pDHSs. Z-scores represent enrichment of observed versus background co-associations computed in 1,000 randomly selected, chromatin-accessible regions.

B   Log2 mRNA expression levels of *Runx1, Ets1,* and *B2m* in untreated and PMA/I-treated T$_N$, T$_M,$ and T$_B$ from the microarray analysis.

    

distribution of AP-1 motifs and ChIP peaks (Fig 5B, Appendix Fig S4). We previously showed above that *Fos* mRNA was induced in $T_N$ by ConA during blast cell transformation (Appendix Fig S1A). These data support the view that AP-1 plays a role in the creation, and subsequent reactivation, but not necessarily the steady-state maintenance of pDHSs when much lower levels of at least *Fos*, *Fosb*, and *Jun* mRNA are detected (Fig EV1A).

A specific example of $T_M$-specific DHSs which contained the predicted motifs and which showed enrichment by ChIP is shown in Fig 5E depicting the transient receptor potential M6 (*Trpm6*) locus which contained multiple $T_M$/$T_B$-specific DHSs. At the −3.2-kb and +21-kb pDHSs, ChIP peaks were detected for RUNX1 and ETS-1 in $T_B$ but not $T_N$. Importantly, motifs for both these factors were present within the hypersensitive region. At the −2.4-kb DHS, we identified a RUNX motif, but no ETS motif, and this was reflected by binding of RUNX1 but not ETS-1. Furthermore, at least 6 DHSs within this locus encompass AP-1 motifs, and this was reflected by specific JUNB binding in $T_B{}^+$ and not in $T_B$ (Fig 5E).

Further efforts are needed to investigate functions of RUNX1 and ETS-1, but this is a difficult issue to address because RUNX and ETS proteins play important roles in many aspects of T-cell development and function (Muthusamy *et al*, 1995; Telfer & Rothenberg, 2001; Egawa *et al*, 2007). We began with a preliminary attempt at evaluating the roles of roles of RUNX1 in blast cell transformation and the activation of inducible cytokine genes associated with pDHSs. For this purpose, we prepared $T_B$ in the presence of either the inhibitor Ro5-3335, which is reported to suppress RUNX1 function (Cunningham *et al*, 2012), or with DMSO as a control. The inhibitor was included during the activation of CD4 T cells by ConA, after which the cells were cultured for 24 h with the inhibitor and IL-2. However, the inhibitor greatly decreased the proportion of T cells that survive the transformation process as live non-apoptotic cells (Appendix Fig S5A). Nevertheless, when the inhibitor was removed and the surviving live cells were stimulated, the residual effect of the inhibitor was to suppress the induction of genes associated with pDHSs (*Il4*, *Il3,* and *CSF2*) but not inducible genes or constitutive genes believed to be independent of pDHSs (*Tnf* and *Cd2*) (Appendix Fig S5B). These data are consistent with a requirement for RUNX1 both in efficient blast cell transformation and in the memory recall response.

## RUNX and ETS motifs are occupied in $T_B$ and $T_M$

To screen for potential additional factors without having to speculate on which antibody to use for ChIP assays, we employed our recently developed Wellington algorithm (Piper *et al*, 2013) to perform *in silico* DNase I footprinting on the DNase-Seq data from $T_B$. This algorithm statistically scores an imbalanced pattern of positive strand reads starting immediately 5′ of the footprint and negative strand reads starting immediately 3′ of the footprint, spanning the binding site which is detected as a region with reduced reads. The distribution of motifs detected by HOMER within these footprints (Fig 7A) resembled that of motifs detected in the full-length DHSs (Fig 5A), with RUNX and ETS the most prominent motifs, accounting for more than half of all the predicted FPs. These were followed by STAT motifs, representing another inducible factor family potentially influencing pDHS formation. Figure 7B depicts the relative DNase I profiles surrounding the FPs at each protected

RUNX, ETS, and STAT motif, showing the relative density of the upper (red) versus lower (green) strand DNase I cuts. Equivalent profiles are shown for footprinted AP-1, KLF, GATA, and RFX motifs in Fig EV4A. FPs were ordered according to increasing occupancy score with the most prominent patterns representing the best FPs (Figs 7B and EV4A). Figures 7B and EV4A also show the average DNase I profiles to illustrate the average cuts on each of the upper (red) and lower (blue) strands centered about the motif. A similar pattern of footprinted RUNX and ETS motifs was detected in $T_M$ (Figs 7C and EV4B). An evaluation of the locations of footprinted ETS and RUNX motifs revealed the same patterns of binding to pDHSs in $T_M$ as was seen for the shared DHSs in $T_N$ (Fig EV4C). Examples of occupied ETS and/or RUNX motifs in $T_B$ are shown for 2 pDHSs located 3.7 and 35 kb upstream of *Ccl1* where binding of both ETS-1 and RUNX1 was demonstrated by ChIP-Seq (Fig 7D). In keeping with a role for the pDHSs in priming inducible gene expression, *Ccl1* is upregulated more efficiently in $T_B$ and $T_M$ compared to $T_N$ (Fig 7E). We also assayed the −3.7-kb and −35-kb pDHSs for potential enhancer activity using the same assay system employed above for testing the *IL3* pDHSs. Similar to the *IL3* pDHSs, these two *Ccl1* pDHSs had no significant enhancer activity in a transfection assay (Fig 7F).

A much smaller percentage of pDHS FPs in non-stimulated $T_B$ (4%) contained occupied AP-1 motifs compared to the number of occupied RUNX and ETS motifs (Fig 7A). However, after stimulation, protection at the AP-1 sites increased significantly in $T_B{}^+$ ($P = 10^{-51}$, Fig 7G), consistent with the increased JUNB binding (Fig 5D) that occurred upon treatment with PMA/I. This was reflected by the distribution of footprint probability scores (Fig 7H).

Taken together, our ChIP and footprinting analyses suggest that RUNX1 and ETS-1 cooperate with inducible factors such as AP-1 to establish pDHS, and are important players in the maintenance of open chromatin at these sites.

## The ratio of inducible versus constitutive transcription factors determines the properties of transient and stable DHS

We next addressed the question of what distinguished constitutive pDHSs from the highly inducible iDHSs in $T_B{}^+$. To this end, we performed a *de novo* motif search using the subset of the 1,217 most enriched iDHSs (Fig 8A). The distribution of motifs in the iDHSs in $T_B{}^+$ was markedly different to the pDHSs in $T_B$ as the pattern was dominated by motifs for the inducible TFs AP-1 (58%) and NFAT (56%). Many of these were composite NFAT/AP-1 motifs of the type first described for the *Il2* promoter and *CSF2* enhancer (Cockerill *et al*, 1995; Chen *et al*, 1998), and observed above in the *Trpm6* +23-kb iDHS (Fig 5E). The full-length composite NFAT/AP-1 motif depicted in Fig 8A was present in 34% of iDHSs, which were also enriched with motifs for the inducible EGR and NF-κB families of TFs. The enrichment of RUNX motifs was considerably lower in iDHSs (12%) than in pDHSs (47%), and ETS motifs were not enriched at all. These results were supported by bootstrapping analyses of the statistical significance of motif co-occurrence within 50 bp, which demonstrated that inducible factors, but not ETS-1 or RUNX1, co-localized in iDHSs (Fig EV5A).

The inducible TF motifs were mapped back onto the combined $T_B$ and $T_B{}^+$ DHS dataset (Fig 8B), once again ranked according to the degree of DHS enrichment in $T_B{}^+$ compared to $T_B$ as in Fig 4A.

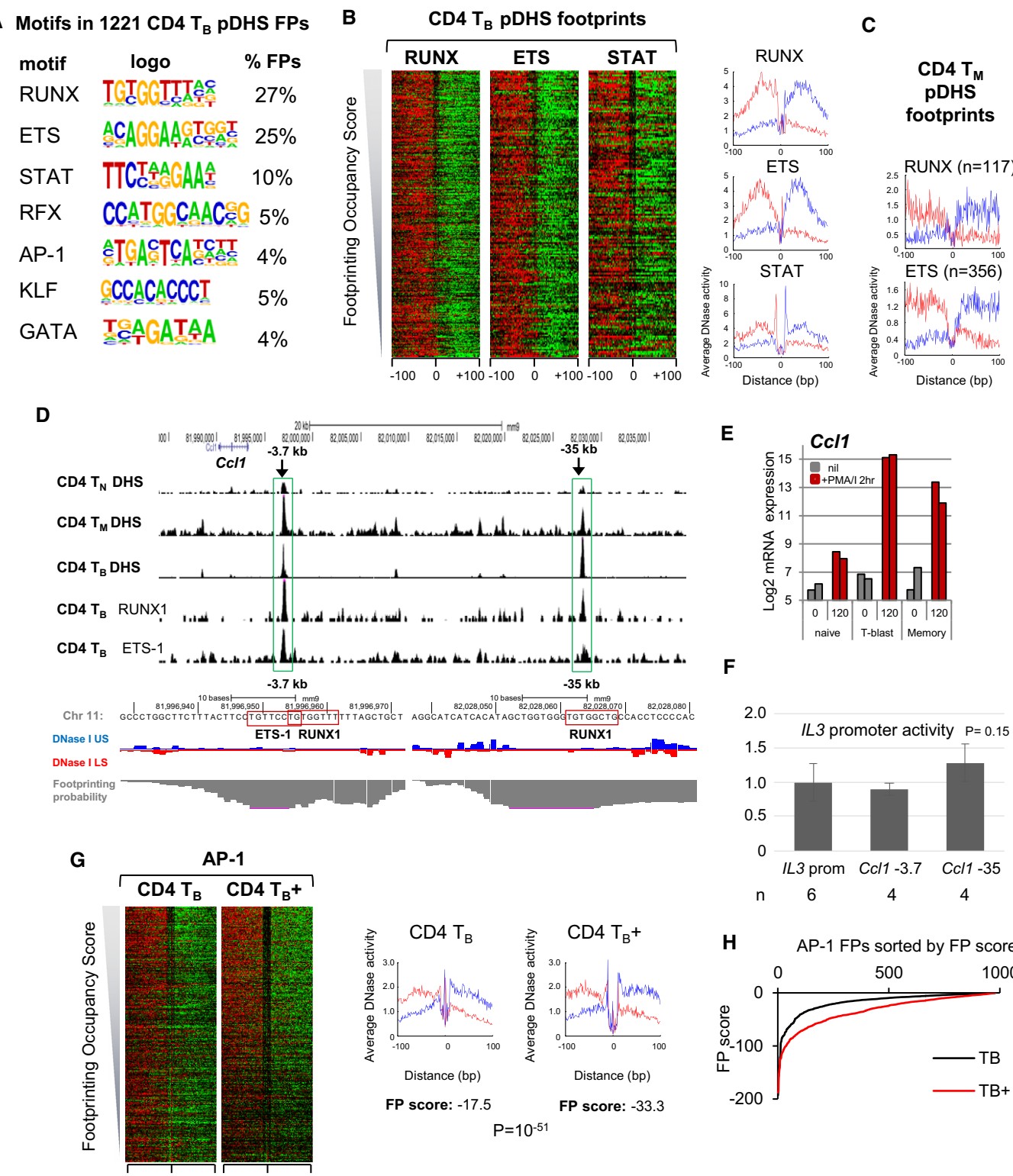

**Figure 7.**

This revealed that the inducible TF binding sites were highly enriched in the iDHSs and were located predominantly at their centers. In contrast, the RUNX motifs and the RUNX1-bound DHSs were divided between the shared DHSs and the iDHSs (Fig 8B and

C), similar to what was observed in the $T_B$ and $T_N$ analysis (Fig 5B). However, the ChIP-Seq analyses showed that RUNX1 only bound to the iDHSs after stimulation (Fig 8C). JUNB ChIP-Seq also confirmed that AP-1 bound specifically to the inducible sites only when the

  

**Figure 7.  Motifs for constitutive TFs are footprinted *in vivo* in pDHSs.**

A   Enriched motifs defined by HOMER using a *de novo* motif search of the digital DNase I FPs identified in the pDHSs in $T_B$.
B   DNase I cleavage patterns in $T_B$ from the FPs determined by Wellington at the pDHSs centered on the motif named at the top and ordered according to increasing FP occupancy score. Left: Cuts are shown within a 200-bp window with positive (red) and negative (green) strand imbalances in DNase I cuts. Right: Average profiles of the actual DNase I cuts at footprinted motifs within the pDHSs, with upper strand DNA cuts shown in red and lower strand cuts in blue.
C   Average profiles of the DNase I cuts at footprinted motifs within the pDHSs in $T_M$ determined as in (B).
D   Example of FP patterns and motifs at the −3.7-kb and −35-kb pDHSs at the *Ccl1* locus in $T_B$.
E   mRNA array values for *Ccl1* expression.
F   Luciferase reporter gene assays in stimulated Jurkat T cells performed as in Fig 1H of the *IL3* promoter alone or in combination with the *Ccl1* −3.7-kb or −35-kb DHSs, with SD. Values are expressed as the mean with the number of replicates for each (*n*) shown underneath.
G   Footprinting of AP-1 sites in $T_B^+$. DNase I upper and lower strand cleavage patterns were calculated as in (B) (left) plus the average DNase I profiles (right) for all AP-1 motifs within the subset of 2,882 defined pDHSs in $T_B$ cells before and after stimulation, ranked in order of decreasing FP probability score.
H   Distribution of the FP probability scores for the data shown in (G).

cells were stimulated (Fig 8C). In contrast, the dDHSs that were lost upon stimulation did not encompass AP-1 motifs. The inability to recruit AP-1 most likely makes them susceptible to the effects of chromatin remodeling that stems from other nearby DHSs which do bind AP-1.

An example of the above patterns of TF binding is shown for the *Il10* locus (Fig 8D) where RUNX1 and JUNB bound to the −24-kb and −30-kb iDHSs only in $T_B^+$. RUNX1 also bound to the −9-kb and −26-kb pDHSs prior to stimulation, whereas AP-1 only bound to these sites after treatment with PMA/I. A similar pattern of binding was observed at the Th2 cytokine gene locus (Fig EV5B). Wellington FP analyses of the $T_B^+$ DNase-Seq data revealed strong footprints at the iDHSs, with AP-1, NFAT, and EGR motifs being the most abundant occupied motifs (Figs 8E and EV5C and D). This result is consistent with the notion that iDHSs are enriched in motifs occupied by multiple species of inducible TFs, responding to concurrent signals. Examples of AP-1 and AP-1/NFAT FPs are shown here for the −15-kb and −35-kb DHSs at the *Ccl1* locus. In this instance, the −35-kb DHS represented a preexisting DHS which recruited AP-1 and became a broader DHS after stimulation. Overall, our analyses identified a strong trend toward a significant difference in the binding motif composition between pDHSs and iDHSs.

The above studies revealed that the iDHSs and pDHSs bind a common set of inducible and constitutively expressed transcription factors yet exhibit different kinetic behaviors within chromatin with one class of binding sites being maintained but not the other. To further investigate the reason for this difference, we determined the number of motifs per DHS within the most enriched population of each class of DHSs. For this purpose, we compared the most specific subset of 1,217 iDHSs with the 2,882 pDHSs identified in $T_M$. To further validate $T_B$ as a surrogate model for $T_M$, we also analyzed the equivalent subset of the 3,085 most enriched DHSs detected in a comparison of $T_B$ and $T_N$ (Fig 9A and B), as these should give the same pattern as $T_M$. As expected, the $T_M$- and $T_B$-specific DHSs had an essentially identical motif composition, with RUNX, ETS, and STAT motifs being abundant, whereas NFAT and AP-1 motifs were more frequent in iDHSs (Fig 9A). We tabulated the motif counts for each dataset for the 5 most abundant motifs for inducible factors (AP-1, NFAT, EGR, NF-κB, and CREB/ATF) and the 5 most abundant motifs for constitutive factors (ETS, RUNX, KLF, GATA, and E-box) (Fig 9B). The motifs used for this purpose were the ones identified as *de novo* motifs by HOMER above (Dataset EV5). An analysis of the ChIP peaks confirmed that ETS-1 and RUNX1 were each bound to two-third of 2,882 pDHSs (Fig 9C). Overall, these analyses revealed that $T_M$- and $T_B$-specific pDHSs contained three times more

motifs for constitutive than inducible factors and the converse was true for iDHSs with the ratio of inducible versus constitutive factor binding sites being two to one. Moreover, the comparison of co-localizing motifs within pDHSs and iDHSs showed a strong co-localization between inducible and constitutive factor binding sites in pDHSs, but not in iDHSs (Figs 6A and EV5A). These different motif compositions are exemplified by the +9-kb pDHS and the +6.5-kb iDHS at the *Syne3* locus, the Th2 LCR, and the *Cxcr3* locus (Appendix Fig S6). In the *Syne3* +9-kb pDHS, there is also an example of a composite ETS/RUNX motif of the type shown in Fig 5A.

Taken together, these data suggest a model whereby the balance of a common set of factors at specific *cis*-elements determines the kinetic behavior of transcription factor assembly and maintenance, and nucleosome occupancy (Fig 9D). This finding supports a model whereby the acquisition of immunological memory in T cells is driven during blast cell transformation by inducible factors that enable the redistribution of preexisting factors. The key to this model is that the density of constitutive factors bound at pDHSs is sufficient to maintain them in the long term, once they have formed, but is insufficient to initially create these DHSs without additional support from inducible factors. The converse is true of the inducible DHSs, which use a high density of inducible motifs to function as rapid response elements, and cannot be maintained once the inducing stimulus is removed. In addition, there will be many other DHSs which share some of the properties of both pDHSs and iDHSs.

## Discussion

This study significantly advances our understanding of the fundamental molecular mechanisms that underpin the long-term maintenance of stable molecular memory in cells that have previously responded to transient stimuli. In our model, inducible factors activated by the primary response in naïve T cells direct the re-localization of the preexisting factors ETS-1 and RUNX1 to thousands of newly established pDHSs which remain associated with pDHSs and continue to modify chromatin structure after the initial stimulation has ceased. Their binding is likely to be further stabilized due to the fact that the ETS motif identified in Fig 5A is a composite ETS/RUNX motif binding these factors cooperatively (Hollenhorst *et al*, 2009). Based on the functional and correlative studies described here, many of these pDHSs are not typically classical enhancers, but function at close range to maintain a regional state of active chromatin in the absence of stimulation and increase the accessibility of inducible enhancers to the factors that direct the immunological

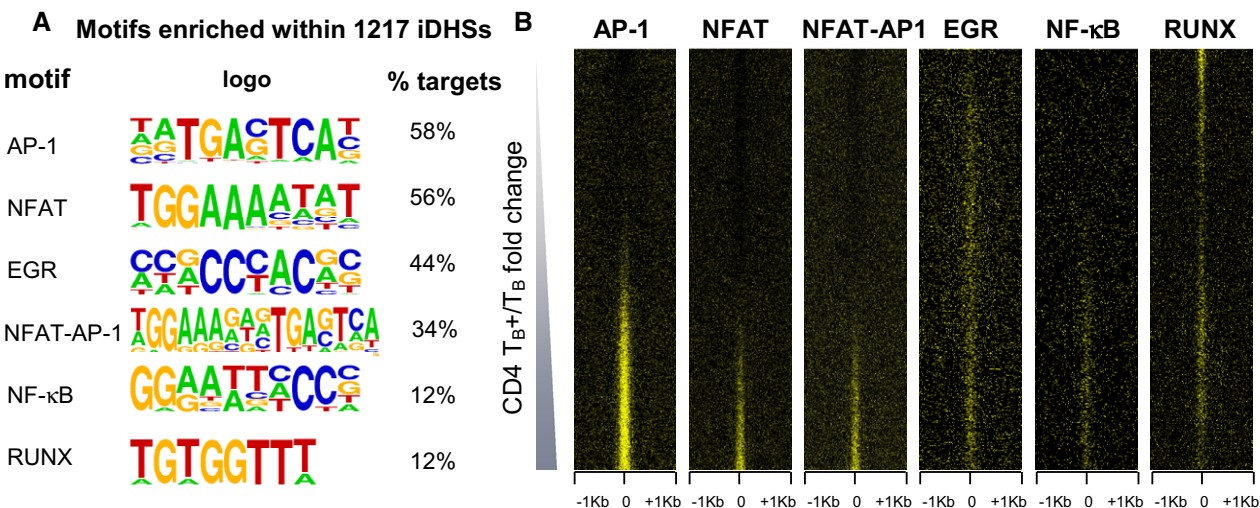

**Figure 8. Inducible DHSs bind inducible and constitutively expressed transcription factors.**

A *De novo* motif search of the 1,217 iDHSs using HOMER.

B Motif distributions in the DHSs ordered by increasing DNase-Seq tag count signal for CD4 $T_B^+$ cells compared to $T_B$ cells as in Fig 4A.

C RUNX1 and JUNB ChIP-Seq density maps depicting binding at all DHSs ordered as in (B).

D RUNX1 and JUNB ChIP-Seq profiles for DHSs at the *Il10* locus in $T_B$ and $T_B^+$.

E DNase I cleavage strand imbalance patterns displayed as in Fig 7G for footprinted TF motifs in $T_B^+$ at the iDHSs.

F Example of FP patterns and motifs at the −15-kb and −35-kb iDHSs in $T_B^+$ cells at the *Ccl1* locus.

recall response. Significantly, the chromatin of inducible enhancers linked to pDHSs in $T_M$ and $T_B$ is inaccessible in $T_N$, and these elements therefore fail to respond efficiently to the same stimuli. In support of the idea of a facilitator role for pDHSs, we demonstrate a tight correlation between the preferential induction of genes in previously activated T cells and the close proximity of (i) pDHSs to iDHSs, and (ii) iDHSs to the TSSs of these inducible genes. Significantly, the presence or absence pDHSs has little impact on steady-state transcription of nearby genes, in contrast to classical enhancers. Others have also recently observed that evidence of TF binding is a much more reliable guide to enhancer function than the mere presence of open chromatin or active histone modifications (Kwasnieski *et al*, 2014; Dogan *et al*, 2015). Given that just 11% of these modified regions tested had strong enhancer activity (Kwasnieski *et al*, 2014), it is evident that there is a growing need to define distinct classes of distal regulatory element.

Our data also provide a valuable resource defining all potential inducible enhancers in T cells, plus the factors required for their activation. The mechanisms driving inducible gene expression in activated T cells have been well defined at numerous individual loci, including the cytokine genes described above. This typically involves the induction of NFAT, AP-1, and NF-κB by TCR signaling pathways, which in turn mediate the induction of DHSs at promoters and enhancers, and the activation of mRNA expression. Here we show that one-third of all iDHSs utilize composite NFAT/AP-1 elements where NFAT and AP-1 are known to bind cooperatively. We previously reported on such a mechanism for the *CSF2* locus (Cockerill *et al*, 1995; Johnson *et al*, 2004), and in the current study, we defined the consensus sequence underpinning the cooperative binding of NFAT and AP-1 for the full complement of iDHSs. Furthermore, our study provides a firm basis for why these enhancers at iDHSs are non-functional in $T_N$, even though $T_N$ efficiently express NFAT and AP-1 upon activation, simply by virtue of their chromatin being inaccessible.

Intriguingly, pDHSs and iDHSs utilize essentially the same set of TFs, yet they behave in very different ways. We provide convincing global evidence that the difference in the ratios of constitutive to inducible TF motifs distinguishes these two classes of regulatory element and accounts for their properties. The low density of inducible TF motifs in pDHSs correlates with a requirement for extensive exposure to TCR signaling during blast cell transformation before the chromatin encompassing these elements is rendered accessible. However, once activated via such a hit-and-run mechanism, their high density of constitutive TF motifs allows them to remain accessible. Conversely, the high density of inducible TF motifs plus low density of constitutive TF motifs in iDHSs ensures their tight regulation whereby they are rapidly induced, but readily become reoccupied by nucleosomes once the stimulus is removed.

Previous studies have found that a subset of DHSs, and a subset of the TFs bound to these DHSs, can be maintained during mitosis (Martinez-Balbas *et al*, 1995; Kadauke *et al*, 2012; Hsiung *et al*, 2015). These studies defined GATA-1 as a mitotic bookmarking factor. However, these studies also found that the majority of mitotically preserved DHSs were localized at promoters, whereas the majority of the distal DHSs which include enhancers were erased. The fact that pDHSs are maintained during multiple rounds of cell division suggests that pDHSs may form a class of distal elements that can, unlike most enhancers, maintain accessible chromatin

during mitosis. This is made more likely by the fact that pDHSs bind co-localizing, constitutively expressed TFs such as RUNX1 and ETS-1 which can bind in a concerted fashion to composite ETS/RUNX elements (Hollenhorst *et al*, 2009). In this context, it is interesting to note that all members of the RUNX transcription factor family have been shown to associate with mitotic chromatin (Young *et al*, 2007; Bakshi *et al*, 2008; Pande *et al*, 2009). Transcription factor complexes recruit chromatin modifiers and chromatin regions flanking pDHSs are marked by the active modifications H3K4me2 and H3K27ac, which can attract chromatin additional modifying complexes containing PHD domain or Bromo domains. We show here that pDHSs bind the co-activator BRD4 whose binding is also maintained during mitosis (Zhao *et al*, 2011). We propose, therefore, that pDHSs contain stable transcription factor complexes that keep such elements nucleosome-free during cell division, thus creating a looser and more dynamic chromatin structure which (i) allows the re-assembly of the full complex after cell division and (ii) exposes more of the chromatin-bound DNA to the inducible TFs searching for their binding sites.

Our current work also increases our understanding of the tightly regulated context-dependent expression of key immunological regulators in differentiated T cells, such as *Il4* in Th2 cells and *Cxcr3* in Th1 cells. When first activated in uncommitted $T_N$, both genes make use of a similar set of TFs to establish specific pDHSs and iDHSs. However, their pDHSs differ with respect to the additional GATA and T-box motifs present, allowing them to exist in an accessible state ready to respond to alternate additional signals. In this way, unpolarized $T_B$ are able to respond to either GATA-3 binding to the Th2 LCR GATA motifs, or TBX21 binding to T-box motifs in the *Cxcr3* −3-kb pDHS, dependent upon which Th2 or Th1 differentiation-inducing signals they subsequently encounter. We propose, therefore, that the acquisition of pDHSs represents a form of chromatin imprinting used universally across all classes of T cells when they are first activated, irrespective of whether they are CD4 or CD8 T cells, and independent of subsequent differentiation decisions. Others have also predicted a role for regions of chromatin marked by H3K4me2 in supporting gene activation in Th1 and Th2 cells (Seumois *et al*, 2014), but the mechanism by which such patterns were established was unclear. The mechanisms described here may also help to account for the origins of a class of preexisting DHSs which are present in activated T cells and recruit FOXP3 during the course of regulatory T-cell differentiation (Samstein *et al*, 2012). Taken together, this body of evidence supports the emerging view that the critical steps in establishing immunological memory do indeed occur early during the initial stages of naïve T-cell activation, and not during the subsequent differentiation stages when additional cell type-specific TFs get recruited (Badovinac *et al*, 2005; Kedzierska *et al*, 2007; Russ *et al*, 2014; Crompton *et al*, 2015).

Last, but not least, our work explains previous studies demonstrating differences in the distribution of activating and repressive chromatin marks in $T_N$ and $T_M$ (Rothenberg & Zhang, 2012) and provides the likely regulatory mechanisms underlying these patterns. A recent study found that a lower level of DNA methylation in the $T_M$ was generally associated with a higher level of induction upon stimulation (Komori *et al*, 2015). There is also evidence linking loss of the repressive H3K27me3 modification with gene activation in effector T cells (Araki *et al*, 2009). However, not all

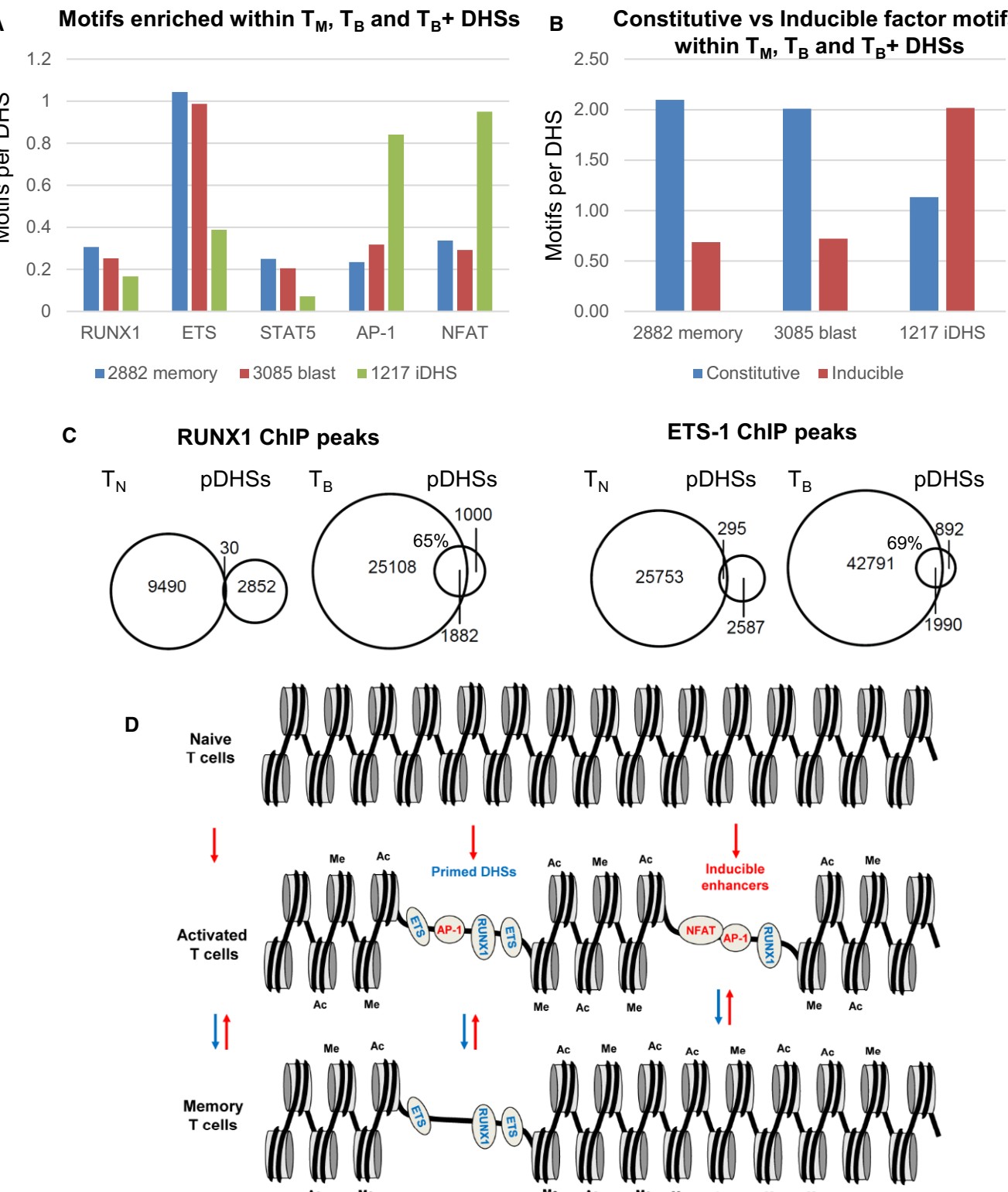

**Figure 9.  Composition and properties of pDHSs and iDHSs.**

A   Motif counts for abundant TF binding sites at the specific DHSs in $T_M$, $T_B$, and $T_B^+$.

B   Total motif counts for 5 inducible motifs (AP-1, NFAT, EGR, NF-κB, and CREB/ATF) and 5 constitutive motifs (ETS, RUNX, KLF, GATA, and E-box) in the specific DHSs in $T_M$, $T_B$, and $T_B^+$. The motifs used here are defined in Dataset EV5.

C   Overlaps between ETS-1 and RUNX1 ChIP peaks and the 2,882 pDHSs in $T_N$ compared to $T_B$.

D   Mechanisms of pDHS and iDHS regulation in T cells.

inducible $T_M$-specific genes are marked by higher levels of H3K27me3 or DNA methylation in $T_N$ prior to TCR activation (Mirabella *et al*, 2010; Zediak *et al*, 2011). These modifications cannot, therefore, be solely responsible for the silent gene state in the naive T cells. Our study now suggests that there are at least two parallel modes of regulation controlling the establishment of the gene expression program in memory T cells: one involving the loss of H3K27me3 and gain of H3K4me3, primarily at promoters, and an additional one involving the *de novo* activation of a separate set of predominantly distal regulatory elements associated with H3K4me2. In the case of inducible genes, our data suggest that it is not primarily loss of repressive modifications in naïve cells that accounts for the memory recall effect in memory T cells. Rather, it is the intrinsic inaccessibility of inactive compact chromatin in $T_N$ and the gain of active modifications to increase accessibility in $T_B$ and $T_N$ that makes gene reactivation a much more efficient process. We present this here as a fundamental mechanism that has the potential to ensure the tight regulation of the immune system (Fig 9D), which is likely to play a major role in the acquisition of immunological memory in T cells and which places the accessibility of chromatin to the binding of transcriptional regulators into the heart of this process.

# Materials and Methods

Detailed methods are available in the Supplementary Methods in the Appendix.

### Mice

C42 transgenic mice containing a 130-kb AgeI genomic DNA fragment of the human *IL3/CSF2* locus on a C57BL/6J background were described previously (Mirabella *et al*, 2010). Studies involving these mice were approved by the ethically reviewed U.K. Home Office animal license PPL 40/3086.

### Cell culture and purification

$T_N$ and $T_M$ were isolated and purified using MACS (Miltenyi) and Easy Sep (Stem Cell Technologies) purification kits (see Appendix Supplementary Methods). $T_B$ were generated from purified $T_N$ cultured at $5 \times 10^5$ cells/ml in IMDM and activated with 2 μg/ml concanavalin A for 40 h. Cells were maintained at $5 \times 10^5$ cells/ml in 50 U/ml IL-2 (Peprotech) for an additional 2–3 days before harvesting. Cells were stimulated with 20 ng/ml phorbol myristate acetate (PMA) and 2 μM calcium ionophore (I) for up to 4 h.

### DNase I-hypersensitive site analysis

DNase I digestion assays were carried out as previously described (Bert *et al*, 2007). A more detailed protocol is available in the Appendix Supplementary Methods.

### Chromatin immunoprecipitation

ChIP for the H3K4me2 and H3K27ac antibodies was carried out as previously described (Lichtinger *et al*, 2012). For all other antibodies,

chromatin was prepared and immunoprecipitated as described in the Supplementary Methods in the Appendix.

### mRNA expression analyses

Total RNA was extracted from cells using TRIzol (Invitrogen). RNA was either (i) reverse-transcribed to cDNA using M-MLV Reverse Transcriptase (Invitrogen) for analysis by quantitative PCR or (ii) prepared for microarray analysis (Supplementary Methods in the Appendix).

### Library preparation

DNA libraries for sequencing were prepared from ~10 ng DNA from DNase I or ChIP samples. Libraries for DNase-Seq of untreated CD4 and CD8 $T_N$ and $T_B$ were prepared for sequencing on the SOLiD™ platform according to the manufacturer's instructions (Applied Biosystems). All other libraries were prepared using the Tru-Seq library preparation kit according to the manufacturer's instructions (Illumina).

### Bioinformatics and data analysis

Detailed data analysis is provided in the Supplementary Methods in the Appendix.

### Accession numbers

ChIP-Seq, DNase-Seq, and expression microarray datasets have been deposited on the Gene Expression Omnibus (GEO) as a super series under GEO accession number GSE67465. Individual datasets were deposited as follows: ChIP-Seq under accession number GSE67443, DNase-Seq under accession number GSE67451, and expression microarrays under accession number GSE67464.

### Public datasets

H3K27me3 ChIP-Seq datasets in naïve CD4 T cells [(Wei *et al*, 2009), GEO accession number GSM361998], naïve, effector, memory CD8 T cells [(Russ *et al*, 2014), SRA accession numbers SRX793474, SRX793485, respectively], Th2 cells [(Wei *et al*, 2009), GEO accession number GSM362002], as well as ENCODE Jurkat DNaseI-Seq [(Thurman *et al*, 2012), GEO accession number GSM736501], and H3K27ac in naïve CD4 T cells [(Lara-Astiaso *et al*, 2014), GEO accession number GSM1441281] were retrieved, aligned, and processed as described above and selected for viewing using the UCSC Genome Browser.

**Expanded View** for this article is available online.

## Acknowledgements

This work was supported by the BBSRC and Bloodwise. We thank Steve Kissane for help with DNA sequencing and micro-array analyses, Maria Rosaria Imperato for help with mRNA analyses, Matthew MacKenzie for help with cell sorting, and Andrew Bert, Fabio Mirabella, Euan Baxter, Sally James, Aude-Marine Bonavita, and Sarion Bowers for assistance with the large body of experimental work leading up to this study. We thank Peter Lane and David Withers for their valuable advice relating to the properties of memory T cells.

## Author contributions

SLB, EB, RCJ, LNG, and PNC performed the laboratory studies, PC, JP, NL, and SO performed bioinformatics analyses, and SLB, PC, CB, and PNC designed the experiments and wrote the manuscript.

## Conflict of interest

The authors declare that they have no conflict of interest.

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
