## [Review Process File · The EMBO Journal]

Manuscript EMBO-2015-92534

Inducible chromatin priming is associated with the establishment of immunological memory in T cells

Sarah L Bevington, Pierre Cauchy, Jason Piper, Elisabeth Bertrand, Naveen Lalli, Rebecca C Jarvis, L. Niall Gilding, Sascha Ott, Constanze Bonifer and Peter N Cockerill

Corresponding author: Peter Cockerill, University of Birmingham

Review timeline:

Submission date:	13 July 2015
Editorial Decision:	12 August 2015
Revision received:	13 November 2015
Editorial Decision:	16 December 2015
Revision received:	17 December 2015
Accepted:	22 December 2015

Editor: Karin Dumstrei

Transaction Report:

1st Editorial Decision

12 August 2015

Thank you for submitting your manuscript to The EMBO Journal. Your study has now been seen by three referees and their comments are provided below.

As you can see, the referees find the analysis interesting and appreciate the extensive dataset. However they also find that further follow up analysis is needed in particular to provide support for that the memory modules are important for memory function. Should you be able to extend the findings along the lines suggested by the referees then we would like to consider a revised version. I should add that it is EMBO Journal policy to allow only a single major round of revision and that it is therefor important to address the raised concerns at this stage.

When preparing your letter of response to the referees' comments, please bear in mind that this will form part of the Review Process File, and will therefore be available online to the community. For more details on our Transparent Editorial Process, please visit our website: http://emboj.embopress.org/about#Transparent_Process

Thank you for the opportunity to consider your work for publication. I look forward to your revision.

REFEREE REPORTS

Referee #1:

The data presented are generally of good quality and potentially interesting. The topic is interesting and the authors have many years of experience with the experimental system.

The main problem with the paper as it stands is how the data are interpreted. It has been known for many years that DHSs can be formed by T cell activation, and that some of these newly formed DHSs persist even after activation. It would be prudent for the authors to acknowledge this fact and to present their data in a way that builds on the existing knowledge.

Current claims of a molecular basis for immunological memory appear overstated:

First, the study does not focus on bona fide memory T cells.

Second, the description of activation-induced DHSs that persist after activation as 'memory modules' is not supported by the current data and should be changed.

Third, there is no experimental evidence that the features described in this study form the basis of accelerated re-activation of inducible genes.

However, the study has major strengths that should form the focus of the presentation:

- (i) Genome-wide mapping of activation-induced DHSs.
- (ii) Thousands of activation-induced DHSs persist after activation.
- (iii) Unbiased identification of activation-induced DHSs that persist after activation allows a much fuller characterisation of such sites than had previously been possible.
- (iv) These sites have interesting features, including composite ETS-1/RUNX binding in T cell blasts, with activation-induced recruitment of AP-1, location near inducible genes (this should be tested statistically, see below).
- (v) There are initial hints at function (some have enhancer function, others cooperate with enhancers), but this analysis should be extended to a reasonable number of sites (see below).

Additional support is required for some of the conclusions:

- Chromatin features of 'memory DHSs': H3K27ac data for naive T cells is required
- Location of activation-induced DHSs that persist after activation: compare to other sites of matched number and size to test enrichment near inducible genes (i) random sites (ii) constitutive DHSs (iii) DHSs that require acute activation
- The authors suggest that the presence of 2882 'memory DHSs' has little impact on baseline levels of gene expression (p12). If this conclusion is based on only the top 1% of differentially expressed genes it requires additional support.
- Throughout the manuscript the authors suggest that 'memory DHSs' are not classical enhancers. This conclusion is central to the manuscript. To support it, the authors should perform additional enhancer assays similar to FigS1D. These data should be shown in one of the main figures rather than in supplementary information.
- The authors say that the role of 'memory DHSs' is to regulate inducible gene expression, but no experimental support is provided for this assertion.
- The authors link 'memory DHSs' to the binding of RUNX and ETS factors. The data provide clear support that RUNX and ETS factors bind 'memory DHSs' in T cell blasts and not in naive T cells. However, an important question that remains to be addressed is whether memory T cells show persistent binding of RUNX and ETS factors at 'memory DHSs'.
- The authors suggest that 'memory DHSs' are maintained by the binding of RUNX and ETS factors.

To support this suggestion it is important to know the fraction of 'memory DHSs' and 'inducible DHSs' that are bound by RUNX and ETS factors in T cell blasts and in T cell blasts after acute stimulation.

Admittedly, even after these points have been addressed, there would still be no evidence that the sites described here are functionally relevant for the re-activation of inducible genes in T cells. Nevertheless, this could be addressed by toning down the narrative. The resulting manuscript would hopefully be of interest to the general readership of the journal, and a significant contribution to the field.

Minor points

- The title of Figure 2 is not informative: the figure presents 'genome-wide mapping of memory-specific DHSs' or similar
- The title of Figure 3 is not informative: the figure suggests that 'memory-specific DHSs are near inducible genes' or similar.
- Figure 7 lacks a title.
- Why is an additional class of 'blast' motifs introduced in Fig 7?
- Depiction of 'closed chromatin': what is the underlying model for the nucleosome arrangement shown? A 'beads on a string' representation would be less contentious and DHSs could be indicated by gaps between nucleosomes. The chromatin cartoons appear to show DNA wrapping nucleosomes in right-handed turns, which would be incorrect.
- The depiction of gene expression changes between naive and 'memory' T cells in Fig 2F suggests 'downregulation' in naive T cells. This should be turned around to show higher inducibility in 'memory' T cells
- The graphical abstract refers to primary and secondary 'infection'. This should be changed to 'activation' as no infection models are used in the manuscript.

Referee #2:

This paper explores an interesting area in immunology, i.e. the mechanism by which T-cells maintain their transcriptional memory. The report uses genome wide analyses of DNase accessibility and CHIP seq for a number of transcription factors in naïve, stimulated, and memory T-cells. Based on the results the authors propose a very tantalizing model in which the type and density of binding sites for select nuclear factors determine whether a DNase hypersensitive site is initiated, stably maintained or lost. Because of their tendency to reside close to inducible enhancer elements, these stable sites are speculated to facilitate rapid activation of enhancer function. Overall the paper is well written and the data are presented nicely. The datasets are of good quality and useful to the field. A limitation of the study is that experimental perturbations to directly test the putative memory functions for the so-called memory modules are lacking. Therefore, the title and the abstract are somewhat overstated.

Major issues:

- Experimental setup. It is not clear why they use a mouse with a human BAC transgene. It is difficult to interpret assertions about conservation of DNase-HS sites between mouse and human given the human BAC is in a mouse trans environment. The presence of the transgene is not used in any particular way, for example to facilitate engineering of mutations at relevant sites. Next, they activate T cells with concanavalin A *ex vivo*, which is a less physiological choice than anti-CD3 anti-CD28 or antigen stimulation, or indeed *in vivo* generation via infection/immunization. The T cells they examine are not TCR transgenic or antigen specific. It would be helpful if the authors explained what physiological population is reflected in their "T-blast" cells. They don't appear fully activated even though they had been stimulated a few days earlier. Does this state occur during an infection *in vivo*? Finally, the memory cells are derived from naïve mouse spleens, leaving it unclear what these memory cells are specific to, or when in the life of the mouse they might have been

generated.

- Genome-wide data analysis: The results should contain a short description of the DNase seq experiments, including total number of reads in each sample, how data were normalized, how peaks were defined and called. The figures should contain scales. It appears that read densities vary among populations. For example, in fig 1D CD4TM the "grass" seems higher. Is this due to more reads or a different scale? It is mentioned in the results that only 83% of the MM DHS are present in a biological replicate DNase-seq sample. The reproducibility of the replicates should be shown, and it would be preferable if called peaks are present in both replicates.
- Biological claims: Some of the claims by authors remain unsupported: for example, in the discussion the authors suggest that MM DHSs are not typical enhancers but a 'novel class of regulatory elements'. Just because a regulatory element does not function in the context of a particular promoter-reporter construct does not necessarily imply a categorically different entity. The term 'memory module' is useful but should be reserved to elements that have been experimentally shown to exert a memory function. For example, does mutation of this element prevent formation of a nearby iDHS and gene activation? The claim that the MMs are mitotically stable is also not supported. The element might disappear during mitosis and then reappear.
- Experimentation: Few of the genome-wide data were validated independently. Importantly, the attractive model proposed by the authors could be strengthened by molecular perturbations. For example, what is the impact of knocking down or out ETS1 and Runx1 on transcriptional memory? Could some of the MM DHS be perturbed to examine how transcription is impacted or whether iDHS are now less accessible?

Additional points:

- Of the MM DHS's, 12% appear to be at promoters while the rest appears to be mostly distal regulatory elements. It is possible that these two groups are functionally distinct and therefore should be analyzed separately.
- Assigning enhancers to promoters is challenging. Recently, an algorithm was published by Kai Tan's group that might be worth trying (He et al., PNAS 111(21) 2191, 2014).
- Most figures show density plots of DNase peaks, which are then analyzed for intensity gains upon activation/memory formation. It would be helpful to identify these peaks on the original density plot and specify how a increased peak is defined, given that the majority of peaks seem to be somewhat present before activation
- What is the reason for focusing on ETS1 and JunB and excluding other family members when considering the ETS and AP-1 motifs in figures 4a and 6a? Why not other members of those families? Please explain.
- What are the known roles of Runx and ETS in T cell biology? Please elaborate.
- It appears that of 1895 memory specific genes induced, only 683 have an MM DHS near them. Do the others have a preexisting DHS nearby?
- Some DNase HS sites are lost upon activation/memory formation? Are these relevant?
- In the DNase tracks shown, the memory cells seem to have a higher background. Why?
- How were footprinting scores determined? Is the FP score sensitive to local DNase signal intensities and if so how is it affected by the DNase peak intensity? Please explain the Wellington algorithm. Also, it would be helpful to zoom in on the footprints in figures 5b and 5c to see the protected nucleotides/motifs more clearly. I
- In graphs like 3e and 3g, it might be more informative to do a scatterplot of increase in gene expression vs kb (from iDHS, for example)- to better illustrate the trend.
- Introduction mentions GATA-2 instead of GATA-3 as a Th2 specific factor
- The authors could at least speculate how putative MM elements function over considerable distances to facilitate DHS formation at another site.
- In light of the above limitations, the title and the final sentence of the abstract overstate what the data are able to tell us.

Referee #3:

This is an elegant study that addresses the molecular basis for the changes in gene expression that are associated with immunological memory. By global analysis of the development of DNase HSs in stimulated or unstimulated naïve, blasting and memory T cells, the authors develop a model for memory that is dependent upon two types of sites: memory module (MM) DHSs (absent in naïve but shared between unstimulated blasting and memory T cells) and inducible (i)DHSs (present only

in restimulated blasting and memory T cells). They show that MM DHSs tend to bind ETS-1 and RUNX1 and tend to be located in the vicinity of iDHSs and inducible genes, but do not themselves exhibit classical enhancer activity. iDHSs, on the other hand, tend to be enriched for binding of inducible transcription factors, such as NFAT and AP1. Based on the hierarchical formation of MM DHSs and iDHSs and the mitotic stability of MM DHSs, the authors hypothesize that MM DHSs are key elements that establish memory (defined by rapid induction of iDHSs and associated gene expression) by priming the formation of nearby iDHSs. Another key proposal is that the formation, but not the maintenance, of MM DHSs depends on inducible transcription factors in T blast cells that set the stage for stable binding of RUNX1 and ETS-1.

A weakness of the paper is that is basically descriptive - guilt by association. What the authors do not do is actually prove their hypothesis, for example, by eliminating MM DHSs from their IL3/CSF2 BAC transgene. However, they do present a fairly thorough and compelling analysis (with some exceptions outlined below), and I do think this work stands on its own and will provide valuable raw material for further mechanistic studies.

Major concerns:

1. The conclusion about the genesis of the MM DHSs seems not well-founded. The authors argue on pg 16 that RUNX1 and ETS-1 are insufficient by themselves to explain occupancy in T blasts but not T naive, pointing to a role for inducible factors. However the argument that AP-1 is important depends on detecting AP-1 binding not in 40 hr T blasts, but in re-stimulated T blasts. AP-1 presumably does not bind to these sites in either resting or stimulated T naïve cells. So the binding that they detect is after-the-fact, and it is not clear that there is any data supporting the notion that AP-1 plays a role in the creation of these sites. It would be useful to conduct a time-course experiment examining the kinetics of DHS formation versus AP-1, RUNX1 and ETS-1 binding in stimulated T cells between the 4 hr (naïve stimulated) and 40 hr (blast) timepoints.
2. The nature of the memory cells that have been analyzed in this study needs to be explicitly discussed. The authors do not provide details, but the EASYSEP Kit that they used should give a mix of effector and central memory T cells that is dominated by effectors. Do their results apply to both effector and central memory T cells or to effector T cells only?
3. Pg 9 and Figure S1E: The authors measure NFAT and AP1 family member transcripts and conclude that naïve and blasting T cells express similar levels of these transcripts before stimulation and after stimulation. From this they conclude that transcription factor expression is not what distinguishes blasts from naïve cells. I am concerned that the authors are oversimplifying the data and I note that there are no statistics applied to the data to evaluate the significance of apparent differences between naïve and blast. Second, the conclusion ignores potential post-transcriptional/translational regulation of these factors. The authors may want to state their conclusion more cautiously. Finally, for the sake of clarity, on pg 9 it would be appropriate to say "distinguishes the responses of TB and TM from TN..."
4. Pg 16: The authors refer to Fig S1E to indicate that increased binding of AP-1 to MM sites in stimulated T blasts was paralleled by upregulation of AP-1 family mRNA levels in those cells. They should point out, however, that upregulation of AP-1 family mRNA levels occurs also in stimulated naïve cells. Thus this upregulation cannot explain in any simple way either binding site occupancy or MM DHS formation.

Minor points:

1. Pg 5: "These DHSs, which we define as Memory Module DHSs, were bound by ETS-1 and RUNX1..." It would be more appropriate to say "Many of these DHSs...". The data are not consistent with ETS/RUNX being universally bound to MM DHSs.
2. Pg 10: "a reproducible subset 2882 of the ...". Fix wording.
3. Pg 11: "H3K27ac only in TB and not in TN..." Where is the data for TN? It is not in 2B or 2E, although stated as such.
4. Pg 12: NFATc1 is identified as a gene that by microarray is upregulated by PMA/I in TB and TM but not TN. But NFATc1 is shown in Fig. S1 to be indistinguishably activated in TN and TB. The authors should address this apparent discrepancy.
5. Fig. 3E: The authors should include statistics: SD, p value.
6. Fig. 3H: Please label the genes.
7. Pg 15: third line, should be Fig. S3C.
8. Pg 16: "may be required to for". Please correct.
9. Pg 17: "was similar to that observed for TB (Fig. S5B)". This should be TM.

10. Pg. 19 3rd line: "MM DHSs prior stimulation..." Please fix.

1st Revision - authors' response

13 November 2015

Referee #1:

The data presented are generally of good quality and potentially interesting. The topic is interesting and the authors have many years of experience with the experimental system.

The main problem with the paper as it stands is how the data are interpreted. It has been known for many years that DHSs can be formed by T cell activation, and that some of these newly formed DHSs persist even after activation. It would be prudent for the authors to acknowledge this fact and to present their data in a way that builds on the existing knowledge.

We have now expanded the introduction to better describe previous studies, including our own, which have previously identified a class of DHS of unknown function that are created in response to T cell activation but prior to terminal differentiation to Th1, Th2 or Treg. We have built upon our studies by deleting one of these elements from its natural location, and made this the starting point of the study.

Current claims of a molecular basis for immunological memory appear overstated:

First, the study does not focus on bona fide memory T cells.

Second, the description of activation-induced DHSs that persist after activation as 'memory modules' is not supported by the current data and should be changed.

Third, there is no experimental evidence that the features described in this study form the basis of accelerated re-activation of inducible genes.

As outlined above, we have now demonstrated that the mouse memory T cells purified by us are CD4+, CD44+, CD62L-ve and CD25-ve (appendix figure 1). This confirms that they have the phenotypic markers characteristic of effector memory T cells, and not activated T cells. However, because we have not used an antigen to immunise the mice, and we have not purified antigen-specific T cells, we have not formally proved that we are working with genuine long term memory T cells. For these reasons we now refer to these cells as "memory-phenotype cells", and we renamed the memory modules as primed DHSs.

We would also like to stress that we did confirm our findings using human peripheral blood memory T cells purified as CD4+ CD45RA-ve cells (Fig 1F). That means we have 2 independent sources of data that support our view that primed DHSs are a characteristic feature of cells which most researchers would regard as some form of memory T cells. For the purpose of this study it is not essential to know just what type of memory cells these are because we are defining general mechanisms that may be shared by all classes of activated and memory T cells.

We believe that the current manuscript does have sufficient data to support the claim that pDHSs are formed during an activation process, and that they are maintained in memory-phenotype cells. The models used in this study are by their very definition previously activated cells, and we now confirm that inducible genes are activated by ConA during blast cell transformation (Appendix Fig S1).

As described above, the revised manuscript does now have data demonstrating accelerated activation of genes in the presence of pDHSs (Fig 2B).

However, the study has major strengths that should form the focus of the presentation:

(i) Genome-wide mapping of activation-induced DHSs.

(ii) Thousands of activation-induced DHSs persist after activation.

(iii) Unbiased identification of activation-induced DHSs that persist after activation allows a much fuller characterisation of such sites than had previously been possible. (iv) These sites have interesting features, including composite ETS-1/RUNX binding in T cell blasts, with activation-induced recruitment of AP-1, location near inducible genes (this should be tested statistically, see below).

(v) There are initial hints at function (some have enhancer function, others cooperate with enhancers), but this analysis should be extended to a reasonable number of sites (see below).

Additional support is required for some of the conclusions:

- Chromatin features of 'memory DHSs': H3K27ac data for naive T cells is required

We have now included a published data set for H3K27ac in Figs 3B and 3E.

- Location of activation-induced DHSs that persist after activation: compare to other sites of matched number and size to test enrichment near inducible genes (i) random sites (ii) constitutive DHSs (iii) DHSs that require acute activation

We addressed this point by computing the number of inducible genes harbouring these populations within 150 kb of the TSS. To compare those with our initial population of 2882 pDHSs, we thus used populations of (i) 2882 randomly generated coordinates, (ii) 2882 randomly selected invariant DHSs (shared in T_N , T_B and T_M), (iii) 2882 randomly selected iDHSs (from a total population of 6823). The number of these sites located within 150 kb of the TSS of an inducible was thus (i) 214/2882, (ii) 223/2882 and (iii) 438/2882. This means that the frequency of random loci, or invariant DHSs is substantially lower than the 683 pDHSs detected within 150 kb, and is also less than the number of pDHSs detected within just 25 kb of inducible genes.

- The authors suggest that the presence of 2882 'memory DHSs' has little impact on baseline levels of gene expression (p12). If this conclusion is based on only the top 1% of differentially expressed genes it requires additional support.

We have included extra analyses comparing mRNA expression in naive versus memory T cells in Figs 3A and appendix Fig S3. When plotted for each nearest gene, there is no significant difference in the baseline expression patterns for these genes.

- Throughout the manuscript the authors suggest that 'memory DHSs' are not classical enhancers. This conclusion is central to the manuscript. To support it, the authors should perform additional enhancer assays similar to FigS1D. These data should be shown in one of the main figures rather than in supplementary information.

We have included several extra sets to support the view that many pDHSs are not classical enhancers. We added an extra construct with the +22 kb pDHS to the GM-CSF promoter analyses shown in Fig 1G. We included a new panel on the IL-3 locus in Fig 1H which includes the previously described IL-3 -34 and -41 kb pDHS constructs plus a newly prepared construct containing both the IL-3 -1.5 and -4.1 kb pDHSs. To make this model applicable more widely to other inducible loci we tested 2 additional pDHSs from CCL1 which also lacked enhancer activity. This adds up to 8 pDHSs tested by us that lack significant classical enhancer activity, in addition to many published ones.

At the same time, we would like to stress that not all pDHSs will behave in this way. There is no reason why some pDHSs should not behave as enhancers, and some in fact do. The point made here is that chromatin priming represents the only function that we can define for many pDHSs, and that active chromatin modifications by themselves are insufficient to define an element as an enhancer. It demonstrates that despite carrying common chromatin marks, genome regulatory elements are heterogeneous and do not all behave the same way. We therefore rephrased our statement to say that pDHSs are not necessarily classical enhancers. We also cite two papers reinforcing this view. This included the Kwasnieski study which found that just 11% of active chromatin regions had strong enhancer activity.

- The authors say that the role of 'memory DHSs' is to regulate inducible gene expression, but no experimental support is provided for this assertion.

As described above, Fig 2B and 2C now include data showing that loss of a pDHS in the IL-3 locus does impair inducible IL-3 expression.

- The authors link 'memory DHSs' to the binding of RUNX and ETS factors. The data provide clear support that RUNX and ETS factors bind 'memory DHSs' in T cell blasts and not in naive T cells. However, an important question that remains to be addressed is whether memory T cells show persistent binding of RUNX and ETS factors at 'memory DHSs'.

For technical reasons it is very difficult to perform ChIP in memory T cells. In place of ChIP we have used DNase I digital footprinting and the Wellington algorithm to predict occupancy of ETS

and RUNX factors at these motifs. These data suggest that these motifs are occupied in memory-phenotype cells (Figs 7C, EV4B and EV4C).

- The authors suggest that 'memory DHSs' are maintained by the binding of RUNX and ETS factors. To support this suggestion it is important to know the fraction of 'memory DHSs' and 'inducible DHSs' that are bound by RUNX and ETS factors in T cell blasts and in T cell blasts after acute stimulation.

We have provided analyses showing the proportion of pDHSs and iDHSs that have ETS-1, RUNX1 and JUNB ChIP peaks (Appendix fig S4, Fig 9C). This shows that ETS-1 and RUNX1 are each present at 2/3 of all pDHSs.

Admittedly, even after these points have been addressed, there would still be no evidence that the sites described here are functionally relevant for the re-activation of inducible genes in T cells. Nevertheless, this could be addressed by toning down the narrative. The resulting manuscript would hopefully be of interest to the general readership of the journal, and a significant contribution to the field.

We have followed both of these recommendations. As described above, we provided evidence that pDHSs are present in memory phenotype cells, and in at least one case, are required for activation of inducible genes. We have also toned down the narrative so as reflect the correlative nature of much of the data.

Minor points

- The title of Figure 2 is not informative: the figure presents 'genome-wide mapping of memory-specific DHSs' or similar

We renamed this legend (Fig 3) as suggested: Genome-wide mapping identifies a class of DHSs restricted to previously activated T cells.

- The title of Figure 3 is not informative: the figure suggests that 'memory-specific DHSs are near inducible genes' or similar.

We renamed this legend (Fig 4) as suggested: iDHSs lie close to pDHSs and are associated with inducible genes.

- Figure 7 lacks a title.

We wrote a title for the legend for Fig 9: Composition and properties on pDHSs and iDHSs.

- Why is an additional class of 'blast' motifs introduced in Fig 7?

It is important to analyse not just the top group of DHSs defined in non-dividing memory-phenotype cells, but also the top DHSs defined in proliferating blast cells. It is highly significant that the 2 independently defined groups show the same features. We have altered the text to make this point clearer.

- Depiction of 'closed chromatin': what is the underlying model for the nucleosome arrangement shown? A 'beads on a string' representation would be less contentious and DHSs could be indicated by gaps between nucleosomes. The chromatin cartoons appear to show DNA wrapping nucleosomes in right-handed turns, which would be incorrect.

We agree that we unintentionally provided a misleading depiction of nucleosome structure. It is however important to convey some meaningful view of the basic underlying organisation of nucleosomes within chromatin. The more common beads on a string models do not convey the degree of chromatin opening that occurs at DHSs. We have therefore redrawn this model to give a somewhat more accurate depiction of the structure (Fig 9D).

- The depiction of gene expression changes between naive and 'memory' T cells in Fig 2F suggests 'downregulation' in naive T cells. This should be turned around to show higher inducibility in 'memory' T cells

The object of what is now Fig 3F is to define a specific population of memory-specific inducible genes. For this reason, the data is presented in a way that highlights the group 1895 genes depicted in black that have higher inducibility.

- The graphical abstract refers to primary and secondary 'infection'. This should be changed to

'activation' as no infection models are used in the manuscript.

The graphical abstract has been changed accordingly, and we redrew the chromatin model.

Referee #2:

This paper explores an interesting area in immunology, i.e. the mechanism by which T-cells maintain their transcriptional memory. The report uses genome wide analyses of DNase accessibility and CHIP seq for a number of transcription factors in naïve, stimulated, and memory T-cells. Based on the results the authors propose a very tantalizing model in which the type and density of binding sites for select nuclear factors determine whether a DNase hypersensitive site is initiated, stably maintained or lost. Because of their tendency to reside close to inducible enhancer elements, these stable sites are speculated to facilitate rapid activation of enhancer function. Overall the paper is well written and the data are presented nicely. The datasets are of good quality and useful to the field. A limitation of the study is that experimental perturbations to directly test the putative memory functions for the so-called memory modules are lacking. Therefore, the title and the abstract are somewhat overstated.

Because referee 1 also asked us to “tone down the narrative” we have made substantial changes to the paper throughout. We have changed not only the title and abstract, but we try to focus throughout on what we have shown, and toned down some claims. We now propose that our data supports a model where primed DHSs contribute to the establishment and maintenance of chromatin priming in blast cells and in memory-phenotype cells. At the same time, we feel justified in highlighting the significant potential of our work for genuine immunological memory and immunity.

Major issues:

- *Experimental setup. It is not clear why they use a mouse with a human BAC transgene. It is difficult to interpret assertions about conservation of DNase-HS sites between mouse and human given the human BAC is in a mouse trans environment. The presence of the transgene is not used in any particular way, for example to facilitate engineering of mutations at relevant sites.*

We have now re-ordered the way the data is introduced. Because this whole study was based on data from the human IL-3/GM-CSF locus, it was logical to include this in the mouse model. The background to the human locus is now described in more detail in the introduction, so that it is now clear why we chose to continue working on this model. The results section now starts with a description of the transgene in Fig 1, and this is followed by a functional study of the human locus in Fig 2. From the very beginning, the transgene gave us the benefit of being able to validate both the general approach and the generality of the findings in mammals in one experimental setting. This served as a launching pad for all the genome-wide studies in the mouse. From a gene regulation perspective, it makes little difference whether the source of the DNA is mouse or human. Tissue-specific trans-regulatory environments are highly conserved as transcription factor motifs have the same sequences in humans and mice and, as exemplified in multiple transgene studies, the way transgenes behave is encoded in their DNA.

Next, they activate T cells with concanavalin A ex vivo, which is a less physiological choice than anti-CD3 anti-CD28 or antigen stimulation, or indeed in vivo generation via infection/immunization. The T cells they examine are not TCR transgenic or antigen specific.

The purpose of our study was to try to define a pan-T cell global mechanism of regulation that occurs during T cell activation. For this purpose it is better to start with as broad a population of cells as possible. We are not trying to zoom in on ever diminishing sub-populations of T cells. All current transgenic models of T cell memory have the opposite problem in that they are far too specific and may be limited to one small sub-population. For this reason we are using an activation signal that will activate all T cells in the population. We also state in the manuscript that the T cells activated by CD3 and CD28 antibodies in our previous study (Baxter et al) have the same properties as the ones treated with ConA. We tend to use ConA more often than antibodies because it is more efficient.

It would be helpful if the authors explained what physiological population is reflected in their "T-blast" cells. They don't appear fully activated even though they had been stimulated a few days earlier. Does this state occur during an infection in vivo?

We agree that it is difficult trying to find a simple terminology to distinguish between (i) a fully activated T cell, expressing inducible genes, and (ii) a previously activated T blast cell, which is actively proliferating, but no longer actively expresses inducible genes unless the TRC signalling pathways are re-stimulated. We have tried to make these distinctions clearer in the text, but it is difficult to adequately define the exact equivalents in vivo.

When T cells become activated in vivo they give rise to a heterogeneous mixture of cells that include some cells actively engaged with antigen in a lymphoid organ, and some cells that may be actively proliferating, with progeny circulating in search of antigen. Only at some of these stages do these cells express inducible genes. If this was not the case, a cytokine storm would ensue. Cytokine responses need to be highly localised, so not all activated T cells are active all the time. In vivo, some of these cells will resemble activated cells undergoing blast cell transformation, and some cells will resemble the in vitro proliferating cells that expand in the absence of stimulation.

Finally, the memory cells are derived from naïve mouse spleens, leaving it unclear what these memory cells are specific to, or when in the life of the mouse they might have been generated.

Even in the absence of immunisation, mice will naturally accumulate a population of memory T cells. It is not necessary to know the identity of the immunogen to define a cell as a memory-phenotype cell. Our surface profiling suggests that the memory cells we purified are predominantly effector memory cells (Appendix Fig S1B).

• Genome-wide data analysis: The results should contain a short description of the DNase seq experiments, including total number of reads in each sample, how data were normalized, how peaks were defined and called. The figures should contain scales. It appears that read densities vary among populations. For example, in fig 1D CD4TM the "grass" seems higher. Is this due to more reads or a different scale? It is mentioned in the results that only 83% of the MM DHS are present in a biological replicate DNase-seq sample. The reproducibility of the replicates should be shown, and it would be preferable if called peaks are present in both replicates.

The most crucial analyses were replicated, and the methods used to process and analyse the genome-wide data, and the read depths, are described in the appendix which shows where duplicate experiments were performed. The read depth of the experiments does vary, as some analyses require very high read depth to obtain higher confidence data for quantitative analyses, such as the footprinting studies. However, the mapping of DHSs at lower depth is sufficient for supporting experiments. Unfortunately, we cannot always control the background, so this is higher in some experiments than others. Given the large amount of data already included, it is not possible to show every relevant track in every figure. We have however shown replicate tracks in Fig EV1B and in Appendix Figure S2.

It is not necessary to include the scale on each figure as this would give a cluttered appearance. It is however important to know how the scales were set in each case. We therefore define and show the scales at the beginning of the Result part (Appendix Fig S2) and we make the statement that the same scales are used in all subsequent figures. What is important here is that each scale starts at 1, and scales have been set so as to allow viewing of peak heights at equivalent levels by viewing multiple sets of ubiquitous peaks.

• Biological claims: Some of the claims by authors remain unsupported: for example, in the discussion the authors suggest that MM DHSs are not typical enhancers but a 'novel class of regulatory elements'. Just because a regulatory element does not function in the context of a particular promoter-reporter construct does not necessarily imply a categorically different entity.

We tried to stress in the manuscript that DHSs represent a heterogeneous group of elements, and that primed DHSs and inducible DHSs are two parts of a broad spectrum. However, it is also evident from our study that many pDHSs do have properties that are different to many iDHSs. We are not trying to claim that all pDHSs are not enhancers, because some do have enhancer activity in vitro or in vivo. What is significant is that a reasonable proportion of pDHSs, and all 8 pDHSs that we tested in enhancer assays, behave very differently in vitro compared to classical inducible enhancers. What is missing here is a deficiency in terminology that treats all enhancers the same, or uses histone modifications as a basis for defining enhancers in place of functional assays. We hope that our study represents one of many which will in future attempt to assign different functions to different classes of enhancers. We suggest that a chromatin priming enhancer will in some cases be a different entity to a transcription activation enhancer, but that there will also be a large overlap between these two across a broad spectrum.

The term 'memory module' is useful but should be reserved to elements that have been experimentally shown to exert a memory function. For example, does mutation of this element prevent formation of a nearby iDHS and gene activation? The claim that the MMs are mitotically stably is also not supported. The element might disappear during mitosis and then reappear.

As explained above, in the absence of a formal definition of immune cells in our study, we have abandoned the term “Memory Module” in favour of the more generic and meaningful term “Primed DHS”. We also abandoned the term “mitotically stable” in favour of “preserved after replication”, as we have not studied what occurs during mitosis.

We also performed the experiment suggested above and found that deletion of a pDHS perturbed the activation of an inducible DHS located 4 kb away.

• Experimentation: Few of the genome-wide data were validated independently. Importantly, the attractive model proposed by the authors could be strengthened by molecular perturbations. For example, what is the impact of knocking down or out ETS1 and Runx1 on transcriptional memory? Could some of the MM DHS be perturbed to examine how transcription is impacted or whether iDHS are now less accessible?

Due to the large number of data sets, we did not have the resources to replicate all of them. Instead we focussed on replicating the most central mRNA and DNase-Seq analyses. The replicate tracks are shown in Appendix fig 2 and will be publicly available.

As suggested by the reviewer we have perturbed a pDHS using CRISPR (Fig 2) and we have suppressed the memory recall response using an inhibitor of RUNX1 function.

Additional points:

• Of the MM DHS's, 12% appear to be at promoters while the rest appears to be mostly distal regulatory elements. It is possible that these two groups are functionally distinct and therefore should be analyzed separately.

The promoters made up a minor component of the DHSs investigated. Given the number of analyses performed, it was not feasible to split these into two sub-groups.

• Assigning enhancers to promoters is challenging. Recently, an algorithm was published by Kai Tan's group that might be worth trying (He et al., PNAS 111(21) 2191, 2014).

We have where possible used our own approach of linking regulated genes with regulated DHSs. We show several such analyses where we show that pDHSs and iDHSs do indeed lie close to inducible genes. We agree that this is more meaningful than just linking a DHS to the nearest gene, which can show a trend but is not very reliable. The above algorithm is based on RNA-Seq and H3K4me1 data and would be difficult to adapt to our study of mRNA array data.

• Most figures show density plots of DNase peaks, which are then analyzed for intensity gains upon activation/memory formation. It would be helpful to identify these peaks on the original density plot and specify how a increased peak is defined, given that the majority of peaks seem to be somewhat present before activation

We have shown the defined pDHSs in Appendix Fig 2, which shows numerous such sites in the Th2 cytokine gene cluster. However, the defined groups of 2882 pDHSs and 1217 iDHSs are not all inclusive and are best used just for the purpose of defining the properties of representative populations of DHSs. They are not be used as an absolute guide to all iDHSs and all pDHSs as they are part of a broad spectrum, and cannot ever include every peak that resembles a pDHS or iDHS or has some ability to regulate a locus. It is best left to individual researchers to evaluate which peaks behave in a significantly different manner in different cells. The raw mapping data will be publicly available and the community is welcome to study the pDHSs relevant to the regulation of their favourite genes, plus other intermediate peaks that may be of interest.

• What is the reason for focusing on ETS1 and JunB and excluding other family members when considering the ETS and AP-1 motifs in figures 4a and 6a? Why not other members of those families? Please explain.

ETS-1 represents perhaps the most relevant ETS protein for immune regulation. JUNB is a family member that remains expressed in activated T cells at a stable level for longer than e.g. FOS or FOSB which are rapidly down-regulated. We were also limited by which ChIP-grade antibodies we succeed in getting to work.

- *What are the known roles of Runx and ETS in T cell biology? Please elaborate.*

It is beyond the scope or space available in this manuscript to expand on the roles of RUNX and ETS proteins in T cells, but it is well established that they are crucial player in T cell development as they are involved in regulating a large number of T cell specific genes such as the TCRs. We cite a review by Rothenberg on T cells which readers can refer to.

- *It appears that of 1895 memory specific genes induced, only 683 have an MM DHS near them. Do the others have a preexisting DHS nearby?*

We have now provided a calculation showing that 91% of the remaining inducible genes did have another type of pre-existing DHS located within 150 kb.

- *Some DNase HS sites are lost upon activation/memory formation? Are these relevant?*

The reviewer raises an interesting point that will in the future form the basis of another whole study. We are also interested in the sites that are lost, and we have now analysed an additional group of 1049 “diminished DHSs” in Fig 4C. Interestingly, this group includes 249 of the 2882 pDHSs, highlighting the dynamic nature of the active chromatin present at loci harbouring pDHSs. Interestingly, these DHSs lack AP-1, which suggests they are unable to resist remodelling that stems from other nearby DHSs which do bind AP-1. Unfortunately, it is far beyond the scope of the current study to speculate further on the significance of these DHSs. It will be exciting to follow this up.

- *In the DNase tracks shown, the memory cells seem to have a higher background. Why?*

The memory cells are a rare population of cells for which it is difficult to generate the same level of high quality data that we achieved for the blast cells. This was as good as we could accomplish with the available resources.

- *How were footprinting scores determined? Is the FP score sensitive to local DNase signal intensities and if so how is it affected by the DNase peak intensity? Please explain the Wellington algorithm. Also, it would be helpful to zoom in on the footprints in figures 5b and 5c to see the protected nucleotides/motifs more clearly.*

We have expanded the description of the footprinting algorithm, and replaced the panels depicting footprinted loci with ones showing a clear motif underlying the footprint in Figs 7E and 8F

- *In graphs like 3e and 3g, it might be more informative to do a scatterplot of increase in gene expression vs kb (from iDHS, for example)- to better illustrate the trend.*

We experimented with scatter plots, but at the end of the day we found that the graphical format used was better at getting the message across. The scatter plots were harder to interpret.

- *Introduction mentions GATA-2 instead of GATA-3 as a Th2 specific factor*

We corrected this error.

- *The authors could at least speculate how putative MM elements function over considerable distances to facilitate DHS formation at another site.*

We have suggested that primed DHSs function to increase accessibility by creating regions of dynamic active chromatin that engulf nearby inducible enhancers. Our current evidence suggests that pDHSs function over a short range, typically within 25 kb, and not over a long range. We tried to convey this concept in Fig 9D.

- *In light of the above limitations, the title and the final sentence of the abstract overstate what the data are able to tell us.*

As described above, we have toned down and rephrased the title, abstract, descriptions of the data, and the conclusions. However, we are still left with a strong message that pDHSs are likely to play a major role in the creation and maintenance of immunological memory.

Referee #3:

This is an elegant study that addresses the molecular basis for the changes in gene expression that

are associated with immunological memory. By global analysis of the development of DNase HSs in stimulated or unstimulated naïve, blasting and memory T cells, the authors develop a model for memory that is dependent upon two types of sites: memory module (MM) DHSs (absent in naïve but shared between unstimulated blasting and memory T cells) and inducible (i)DHSs (present only in restimulated blasting and memory T cells). They show that MM DHSs tend to bind ETS-1 and RUNX1 and tend to be located in the vicinity of iDHSs and inducible genes, but do not themselves exhibit classical enhancer activity. iDHSs, on the other hand, tend to be enriched for binding of inducible transcription factors, such as NFAT and AP1. Based on the hierarchical formation of MM DHSs and iDHSs and the mitotic stability of MM DHSs, the authors hypothesize that MM DHSs are key elements that establish memory (defined by rapid induction of iDHSs and associated gene expression) by priming the formation of nearby iDHSs. Another key proposal is that the formation, but not the maintenance, of MM DHSs depends on inducible transcription factors in T blast cells that set the stage for stable binding of RUNX1 and ETS-1.

A weakness of the paper is that is basically descriptive - guilty by association. What the authors do not do is actually prove their hypothesis, for example, by eliminating MM DHSs from their IL3/CSF2 BAC transgene. However, they do present a fairly thorough and compelling analysis (with some exceptions outlined below), and I do think this work stands on its own and will provide valuable raw material for further mechanistic studies.

We accept that the original manuscript strongly suggested models of T cell regulation based on correlations without providing formal proof of principle. To address this concern we have provided the studies described above to perturb the priming of inducible genes by deleting a primed DHS from the IL-3 used, and by suppressing RUNX1 function (Figures 2 and 6).

Major concerns:

1. The conclusion about the genesis of the MM DHSs seems not well-founded. The authors argue on pg 16 that RUNX1 and ETS-1 are insufficient by themselves to explain occupancy in T blasts but not T naïve, pointing to a role for inducible factors. However the argument that AP-1 is important depends on detecting AP-1 binding not in 40 hr T blasts, but in re-stimulated T blasts. AP-1 presumably does not bind to these sites in either resting or stimulated T naïve cells. So the binding that they detect is after-the-fact, and it is not clear that there is any data supporting the notion that AP-1 plays a role in the creation of these sites. It would be useful to conduct a time-course experiment examining the kinetics of DHS formation versus AP-1, RUNX1 and ETS-1 binding in stimulated T cells between the 4 hr (naïve stimulated) and 40 hr (blast) timepoints.

It is widely accepted that AP-1 is a tightly regulated family of transcription factors which are induced in response to activation of TCR signalling pathways, and it is evident that AP-1 returns to very low levels when TCR signalling pathways are turned off.

The mechanisms that establish active chromatin priming during T blast cell transformation are of great interest, but are very difficult to unravel in a detailed way. The studies proposed here are technically very difficult to perform because they take place over 24 to 48 hours and may involve very gradual changes brought about by low level chronic signalling. We are unlikely to detect this response by CHIP. The cells within this population are heterogeneous and may not behave in a synchronised way. We currently feel that at the present time this is beyond both the scope of the manuscript, and our current resources.

Nevertheless, we have confirmed that there is an activation process occurring during blast cell transformation that involves AP-1. In Appendix Fig S1A we now show that Fos mRNA is induced within 3 hours of stimulation by ConA, and that the CSF2 and Ccl1 genes which are activated by AP-1 in blast cells are also activated by ConA in naïve T cells, but much more gradually. This supports the notion that primed DHSs associated with inducible genes are being created within a window when the same genes are known to be activated by an AP-1-dependent process.

2. The nature of the memory cells that have been analyzed in this study needs to be explicitly discussed. The authors do not provide details, but the EASYSEP Kit that they used should give a mix of effector and central memory T cells that is dominated by effectors. Do their results apply to both effector and central memory T cells or to effector T cells only?

We agree with the need to better define the populations of T cells used in this study. We have therefore performed FACS profiling of the cells used (Appendix Fig 1B) which shows that: (i) the naïve T cells were predominantly CD62L⁺ cells lacking the activation markers CD44 and CD25, (ii) the memory T cells were predominantly effector memory-phenotype cells expressing CD44 but

lacking CD62L and CD25, and (iii) the blast cells predominantly expressed the activation markers CD25 and CD44, and resemble activated effector T cells.

3. Pg 9 and Figure S1E: The authors measure NFAT and AP1 family member transcripts and conclude that naïve and blasting T cells express similar levels of these transcripts before stimulation and after stimulation. From this they conclude that transcription factor expression is not what distinguishes blasts from naïve cells. I am concerned that the authors are oversimplifying the data and I note that there are no statistics applied to the data to evaluate the significance of apparent differences between naïve and blast. Second, the conclusion ignores potential post-transcriptional/translational regulation of these factors. The authors may want to state their conclusion more cautiously. Finally, for the sake of clarity, on pg 9 it would be appropriate to say "distinguishes the responses of TB and TM from TN..."

We agree that signalling and transcription regulation are highly complex, and that the above issues are difficult to unravel here. For this reason we changed the text as requested so as to refer simply to mRNA values, without overstating the claims about the factors themselves. The kinetics of the various inducible factors varies both within families and between cell types, and we have tried to convey this view.

4. Pg 16: The authors refer to Fig S1E to indicate that increased binding of AP-1 to MM sites in stimulated T blasts was paralleled by upregulation of AP-1 family mRNA levels in those cells. They should point out, however, that upregulation of AP-1 family mRNA levels occurs also in stimulated naïve cells. Thus this upregulation cannot explain in any simple way either binding site occupancy or MM DHS formation.

To address this concern about the discussion of S1E (now EV1A) we have modified the text referred to above so as to include the following 2 sentences: "We also showed above that *Fos* mRNA was induced in T_N by ConA during blast cell transformation (Appendix Fig S1A). These data support the view that AP-1 plays a role in the creation, and subsequent reactivation, but not necessarily the steady state maintenance of MM DHSs when much lower levels of at least *Fos*, *Fosb* and *Jun* mRNA are detected (Fig EV1A)."

Minor points:

1. Pg 5: "These DHSs, which we define as Memory Module DHSs, were bound by ETS-1 and RUNX1..." It would be more appropriate to say "Many of these DHSs...". The data are not consistent with ETS/RUNX being universally bound to MM DHSs.

Modified as requested.

2. Pg 10: "a reproducible subset 2882 of the ...". Fix wording.
Wording fixed.

3. Pg 11: "H3K27ac only in TB and not in TN..." Where is the data for TN? It is not in 2B or 2E, although stated as such.

We have now included published data for H3K27ac in T_N.

4. Pg 12: NFATc1 is identified as a gene that by microarray is upregulated by PMA/I in TB and TM but not TN. But NFATc1 is shown in Fig. S1 to be indistinguishably activated in TN and TB. The authors should address this apparent discrepancy.

This discrepancy arose because NFATc1 mRNA changes were above the threshold in TM but not in TB. We have now included an additional analysis of the cumulative mRNA array data for 2 alternate transcripts of NFATc1 (Fig EV2F) which shows a low level of induction in TN and TB, comparable to the qPCR data in Fig EV1A for a common NFATc1 exon, and a higher induction in TM.

5. Fig. 3E: The authors should include statistics: SD, p value.

We have included P values in what are now Figs 4F and 4H.

6. Fig. 3H: Please label the genes.

We labelled the genes in what is now Fig 4I.

7. Pg 15: third line, should be Fig. S3C.

We rephrased this sentence.

8. Pg 16: "may be required to for". Please correct.

We corrected this error.

9. Pg 17: "was similar to that observed for TB (Fig. S5B)". This should be TM.

We corrected this error.

10. Pg. 19 3rd line: "MM DHSs prior stimulation..." Please fix.

We corrected this error.

2nd Editorial Decision

16 December 2015

Thanks for sending us your revised version. Your manuscript has now been re-reviewed by the three referees and their comments are provided below.

As you can see below, the referees appreciate that the analysis has been strengthened. While referees #1 and 3 raise only a few issues to be sorted out, referee #2 still finds the analysis too descriptive and find that further functional data would be needed to support the key conclusions drawn. However, I don't think that the descriptive nature of the manuscript takes away from the interest and impact of the findings. Given this, I would therefore like to invite you to submit a revised version that addresses the minor comments raised. Once we get the revised manuscript back, I will proceed with the acceptance of the manuscript for publication here.

A few comments on the revisions:

Referee #1: point regarding the Runx inhibitor experiment. I see the referee's point, but also think that removing the data from the manuscript is not a good solution either as I think the findings add to the manuscript. Could we maybe add it as an expanded figure or to the Appendix? Make sure that you have balanced discussion about the findings from this experiment to reflect the referee's concern.

Referee #2 still finds the title is overstated - I am a bit torn about this point - could you maybe use the term associate instead of contribute? We can discuss this point further.

- We labeled the expanded view tables as Dataset EV1 etc this allows them to be displayed as excel files in the final version. Can you make sure that in the manuscript that there are referenced as such in the manuscript tex.t.

- Dataset #5 is an empty file. I guess it should be labeled as Dataset EV5. Can you check and upload again. Please also make sure that there is a referenced to this dataset in the text

- I think the accession numbers should be included in the main manuscript and not as now in the appendix.

- In cases where N =2 like (figure 6D and parts of figure 1 - correct me if I am wrong) it would be nicer if you show the values from both experiments rather than showing SD.

I think that should pretty much be it. Let me know if we need to discuss anything further. You can use the link below to upload the revised version.

REFEREE REPORTS

Referee #1:

I would like to thank the authors for their detailed responses and in particular for toning down the narrative.

In addition, the authors have made an ambitious attempt to make the paper more mechanistic and present a large amount of new experimental data. Some of these data strengthen the paper (e.g. testing additional HSSs for enhancer function) but others are too preliminary, particularly the Runx inhibitor experiments.

The authors added the putative Runx inhibitor Ro-5-3335 during T cell activation, and it killed most of the cells over the 64 hour treatment period. Although the remaining cells showed defective induction of some key genes, the impact of the inhibitor on DHSs was not assessed. If the point of this experiment was to evaluate the contribution of Runx to the activation of DHSs, it might have been better to activate naive T cells in the presence of the inhibitor and measure the impact on DHS formation. The extended period of treatment required to evaluate the contribution of Runx to the maintenance of activation-induced DHSs makes the use of the inhibitor highly problematic. Genetic approaches to knock down or to delete Runx1 after activation would perhaps have been better, followed by an assessment of DHSs and gene expression. As it stands, this experiment is not strong enough to be presented here, at least not in a main figure. I am not sure that experimental support for a role of Runx1 is required for this manuscript.

Referee #2:

This manuscript is somewhat improved mostly due to textual changes with minimal new experimentation.

1) In response to all three reviewers, the authors supposedly toned down claims regarding T cell memory. Yet the manuscript still has this topic front and center, including in the title and the beginning of the introduction. Not a single experiment was carried out to examine the relevance of persistent DNase hypersensitive sites or transcription factors to T cell memory. The title and running title need to be changed.

2) As before the authors make assertions about the functions of arbitrarily selected hypersensitive sites (e.g. +11.5 Fig.4I) that are surrounded by many others. There is no experimental or even hypothetical basis to ascribe any gene specific functions to any them. In this example, several HSSs become more prominent, some less, and it remains unclear what these changes mean functionally.

3) The colors in figure 2 are reversed.

In sum, the data are of good quality with notable exceptions as to biological replicates/reproducibility in several experiments. However, the report remains completely descriptive and contains sections that are quite tedious to read.

Referee #3:

The revised manuscript is significantly improved and is acceptable for publication. I have only a few very minor changes to suggest:

1. pg 8 top, line 4: "like the like the". Please correct.
2. pg 9 bot: "Overall to maintain immunological memory". This has generally been toned down elsewhere but remains perhaps too strong here. I suggest "that may function to prime gene expression upon T cell reactivation" or the like.

2nd Revision - authors' response

17 December 2015

Author's response to the request for minor modifications:

We have complied with all the requests below, and have in addition made some additional improvements to reflect some of the original concerns of the reviewers which may not have been fully addressed. In addition to the responses described below, we have

(a) Included additional references for the functions of RUNX and ETS proteins in T cells because these are required to both support their function throughout T cell development, and also to introduce caution into attempts to interpret RUNX1 inhibitor data.

(b) We included a scatter plot originally requested by reviewer 2 in Appendix Fig S3C because we found that this did very elegantly illustrate the close association of primed DHSs with inducible DHSs, independent of their positions relative to the transcription start site. This further supports a model whereby primed DHSs can function locally by supporting enhancer function.

Comments from the editor:

A few comments on the revisions:

Referee #1: point regarding the Runx inhibitor experiment. I see the referee's point, but also think that removing the data from the manuscript is not a good solution either as I think the findings add to the manuscript. Could we maybe add it as an expanded figure or to the Appendix? Make sure that you have balanced discussion about the findings from this experiment to reflect the referee's concern.

We moved the data to the appendix, and provided a more balanced discussion, as stated below for the response to referee 1.

Referee #2 still finds the title is overstated - I am a bit torn about this point - could you maybe use the term associate instead of contribute? We can discuss this point further.

We changed the title and running title as described below.

- We labeled the expanded view tables as Dataset EV1 etc this allows them to be displayed as excel files in the final version. Can you make sure that in the manuscript that there are referenced as such in the manuscript text.

We have used the term Dataset EV throughout.

- Dataset #5 is an empty file. I guess it should be labeled as Dataset EV5. Can you check and upload again. Please also make sure that there is a referenced to this dataset in the text

- I think the accession numbers should be included in the main manuscript and not as known in the appendix.

We moved the sections describing accession numbers from the appendix to the main text.

- In cases where $N=2$ like (figure 6D and parts of figure 1 - correct me if I am wrong) it would be nicer if you show the values from both experiments rather than showing SD.

We used 2 columns side by side in place of average values for duplicates in graphs throughout.

Referee #1:

I would like to thank the authors for their detailed responses and in particular for toning down the narrative.

In addition, the authors have made an ambitious attempt to make the paper more mechanistic and present a large amount of new experimental data. Some of these data strengthen the paper (e.g. testing additional HSSs for enhancer function) but others are too preliminary, particularly the Runx inhibitor experiments.

The authors added the putative Runx inhibitor Ro-5-3335 during T cell activation, and it killed most of the cells over the 64 hour treatment period. Although the remaining cells showed defective induction of some key genes, the impact of the inhibitor on DHSs was not assessed. If the point of

this experiment was to evaluate the contribution of Runx to the activation of DHSs, it might have been better to activate naive T cells in the presence of the inhibitor and measure the impact on DHS formation. The extended period of treatment required to evaluate the contribution of Runx to the maintenance of activation-induced DHSs makes the use of the inhibitor highly problematic. Genetic approaches to knock down or to delete Runx1 after activation would perhaps have been better, followed by an assessment of DHSs and gene expression. As it stands, this experiment is not strong enough to be presented here, at least not in a main figure. I am not sure that experimental support for a role of Runx1 is required for this manuscript.

We agree that treating T cells with a RUNX1 inhibitor is likely to lead to complications due to the fact that RUNX1 plays many important roles in both T cell development and activation, and is likely to cause toxicity as we have observed. To reflect this we have moved the inhibitor data to the appendix and we have described the data as preliminary, and alluded to the problems of suppressing essential genes, as follows:

“Further efforts are needed to investigate functions of RUNX1 and ETS1, but this is a difficult issue to address because RUNX and ETS proteins play important roles in many aspects of T cell development and function (Egawa et al, 2007; Muthusamy et al, 1995; Telfer & Rothenberg, 2001). We began with a preliminary attempt at evaluating the roles of roles of RUNX1 in blast cell transformation and the activation of inducible cytokine genes associated with pDHSs. For this purpose we prepared TB in the presence of either the inhibitor Ro5-3335, which is reported to suppress RUNX1 function (Cunningham et al, 2012), or with DMSO as a control. The inhibitor was included during the activation of CD4 T cells by ConA, after which the cells were cultured for 24 hours with the inhibitor and IL-2. However, the inhibitor greatly decreased the proportion of T cells that survive the transformation process as live non-apoptotic cells (Appendix Fig S5A). Nevertheless, when the inhibitor was removed and the surviving live cells were stimulated, the residual effect of the inhibitor was to suppress the induction of genes associated with pDHSs (*IL4*, *IL3* and *CSF2*) but not inducible genes or constitutive genes believed to be independent of pDHSs (*Tnf* and *Cd2*) (Appendix Fig 5B). These data are consistent with a requirement for RUNX1 both in efficient blast cell transformation and in the memory recall response.”

Referee #2:

This manuscript is somewhat improved mostly due to textual changes with minimal new experimentation.

1) In response to all three reviewers, the authors supposedly toned down claims regarding T cell memory. Yet the manuscript still has this topic front and center, including in the title and the beginning of the introduction. Not a single experiment was carried out to examine the relevance of persistent DNase hypersensitive sites or transcription factors to T cell memory. The title and running title need to be changed.

We have taken extra measures to present the data in a more cautious manner. We have changed the title to “Inducible chromatin priming is associated with the establishment of immunological memory in T cells.” And the running title to “T cell activation leads to epigenetic priming”

2) As before the authors make assertions about the functions of arbitrarily selected hypersensitive sites (e.g. +11.5 Fig.4I) that are surrounded by many others. There is no experimental or even hypothetical basis to ascribe any gene specific functions to any them. In this example, several HSSs become more prominent, some less, and it remains unclear what these changes mean functionally.

The choice of the IL-4/RAD50 locus was not arbitrary. In the description of Fig 4I we specifically described 2 studies which experimentally showed that the two highlighted inducible DHSs do indeed have enhancer activity.

3) The colors in figure 2 are reversed.

We corrected the colours of the legend in Fig 2B and also the colour of one track in Fig S2

In sum, the data are of good quality with notable exceptions as to biological replicates/reproducibility in several experiments. However, the report remains completely descriptive and contains sections that are quite tedious to read.

We provided Venn diagrams in Fig S2 to show that for the DHS data sets where we have duplicates, the peak detection is very reproducible.

Referee #3:

The revised manuscript is significantly improved and is acceptable for publication. I have only a few very minor changes to suggest:

1. pg 8 top, line 4: "like the like the". Please correct.

This is corrected

2. pg 9 bot: "Overall to maintain immunological memory". This has generally been toned down elsewhere but remains perhaps too strong here. I suggest "that may function to prime gene expression upon T cell reactivation" or the like.

We have reworded this passage so as to be more cautious about what pDHSs may do